# Palmitoylation targets the calcineurin phosphatase to the phosphatidylinositol 4-kinase complex at the plasma membrane

Idil Ulengin-Talkish[1], Matthew A. H. Parson[2], Meredith L. Jenkins[2], Jagoree Roy[1], Alexis Z. L. Shih[3,9], Nicole St-Denis[4,10], Gergo Gulyas[5], Tamas Balla[5], Anne-Claude Gingras[4,6], Péter Várnai[7], Elizabeth Conibear[3], John E. Burke[2,8] & Martha S. Cyert[1✉]

Calcineurin, the conserved protein phosphatase and target of immunosuppressants, is a critical mediator of $Ca^{2+}$ signaling. Here, to discover calcineurin-regulated processes we examined an understudied isoform, CNAβ1. We show that unlike canonical cytosolic calcineurin, CNAβ1 localizes to the plasma membrane and Golgi due to palmitoylation of its divergent C-terminal tail, which is reversed by the ABHD17A depalmitoylase. Palmitoylation targets CNAβ1 to a distinct set of membrane-associated interactors including the phosphatidylinositol 4-kinase (PI4KA) complex containing EFR3B, PI4KA, TTC7B and FAM126A. Hydrogen-deuterium exchange reveals multiple calcineurin-PI4KA complex contacts, including a calcineurin-binding peptide motif in the disordered tail of FAM126A, which we establish as a calcineurin substrate. Calcineurin inhibitors decrease PI4P production during Gq-coupled GPCR signaling, suggesting that calcineurin dephosphorylates and promotes PI4KA complex activity. In sum, this work discovers a calcineurin-regulated signaling pathway which highlights the PI4KA complex as a regulatory target and reveals that dynamic palmitoylation confers unique localization, substrate specificity and regulation to CNAβ1.

[1] Department of Biology, Stanford University, Stanford, CA, USA. [2] Department of Biochemistry and Microbiology, University of Victoria, Victoria, BC, Canada. [3] Department of Medical Genetics, University of British Columbia, Vancouver, Canada. [4] Lunenfeld-Tanenbaum Research Institute at Mount Sinai Hospital, University of Toronto, Toronto, Canada. [5] Section on Molecular Signal Transduction, National Institute of Child Health and Human Development, National Institutes of Health, Bethesda, MD, USA. [6] Department of Molecular Genetics, University of Toronto, Toronto, ON, Canada. [7] Department of Physiology, Faculty of Medicine, Semmelweis University, Budapest, Hungary. [8] Department of Biochemistry, The University of British Columbia, Vancouver, BC, Canada. [9] Present address: Max-Delbrück Center for Molecular Medicine, Berlin, Germany. [10] Present address: High-Fidelity Science Communications, Summerside, PE, Canada. ✉email: mcyert@stanford.edu

Cells respond to changes in their environment via signaling pathways, including those regulated by calcium ions (Ca²⁺). Dynamic changes in the intracellular Ca²⁺ concentration provide specific temporal and spatial cues that direct a myriad of physiological responses. Hence, elucidating mechanisms that initiate Ca²⁺ signaling and identifying downstream Ca²⁺ sensing-effectors are critical for understanding cellular regulation in both healthy and diseased cells.

Calcineurin (CN), also known as PP2B or PPP3, is the conserved Ca²⁺/calmodulin (CaM)-activated serine/threonine protein phosphatase, that transduces Ca²⁺ signals to regulate a wide-array of physiological processes. In humans, CN is ubiquitously expressed and has well-established roles in the cardiovascular, nervous, and immune systems[1–3]. Because CN dephosphorylates Nuclear Factor of Activated T-cells (NFAT) transcription factors to activate the adaptive immune response[4], CN inhibitors FK506 (Tacrolimus), and cyclosporin A (CsA) are in wide clinical use as immunosuppressants[5]. However, by inhibiting CN in non-immune tissues, these drugs also provoke a variety of unwanted effects, underscoring the need to comprehensively map CN signaling throughout the body. Recently, systematic discovery of CN targets revealed that many CN-regulated pathways are yet to be elucidated[6,7]. Here, we uncover aspects of CN signaling by focusing on an understudied isoform, CNAβ1.

Calcineurin is an obligate heterodimer of catalytic (CNA) and regulatory (CNB) subunits. In mammals, three isoforms of CNA (α, β and γ) are encoded by separate genes with tissue-specific expression. These isoforms display a similar architecture containing a catalytic domain, binding sites for CNB and CaM, and a C-terminal autoinhibitory domain (AID) which blocks phosphatase activity under basal conditions. Under elevated Ca²⁺ conditions, Ca²⁺ and Ca²⁺/CaM bind to CNB and CNA, respectively, to disrupt AID binding to the catalytic site[8,9]. This activation mechanism is conserved across all CN isoforms in animals and fungi, with the only known exception being a transcript variant of the CNAβ gene, termed CNAβ1[10–13].

Alternative 3′ end processing of the *PPP3CB* mRNA gives rise to two CNAβ isoforms, CNAβ2 with canonical architecture, and the non-canonical CNAβ1[10,11] (Fig. 1a). CNAβ1 is conserved in vertebrates (Fig. 1b) and broadly expressed in human tissues at a low level, alongside the canonical CN isoforms[11,12]. CNAβ1 and CNAβ2 sequences are identical through the CaM-binding domain, but exclusion of two terminal exons and subsequent translation of intronic sequences results in a divergent hydrophobic C-terminus for CNAβ1 that lacks the AID, but contains a distinct autoinhibitory sequence, ⁴⁶²LAVP⁴⁶⁵, which impedes substrate binding[12]. CN recognizes two short, degenerate peptide motifs, "PxIxIT" and "LxVP", in the disordered regions of its substrates[13]. LxVP motifs bind to a region at the CNA/CNB interface that is accessible only after Ca²⁺/CaM binding[13–15] and is blocked by FK506 and CsA showing that LxVP recognition is essential for dephosphorylation[15]. Notably, the unique LxVP-mediated autoinhibition displayed by CNAβ1 is only partially relieved by Ca²⁺/CaM and limits maximal activity in vitro when compared to CNAβ2[12]. However, regulation of CNAβ1 in vivo remains to be investigated.

The current CN literature is focused on canonical isoforms, however the few published studies about CNAβ1 reveal its unique physiological roles: In mouse cardiomyocytes CNAβ1 over-expression is cardio-protective following myocardial infarction, in contrast to pro-hypertropic CNAβ2[16–18]. Furthermore, mice specifically lacking CNAβ1 are viable, but exhibit cardiac hypertrophy and metabolic alterations[18]. CNAβ1 also regulates mouse embryonic stem cell differentiation and activates mTORC2/AKT signaling through an undetermined mechanism that may be independent of its catalytic activity[11,16,19]. Additionally, unlike CNAβ2, CNAβ1 does not dephosphorylate

NFAT[16], and its direct substrates are yet to be identified. Thus, elucidation of these targets promises to reveal unknown aspects of Ca²⁺ and CN signaling.

Some of the best-characterized Ca²⁺ signaling pathways are initiated by ligand binding to Gq-protein coupled receptors (GPCR), causing phospholipase C (PLC) to hydrolyze phosphatidylinositol 4,5-biphosphate (PI(4,5)P₂ or PIP₂) into diacylglycerol (DAG) and inositol triphosphate (IP₃), which activate protein kinase C (PKC) and intracellular Ca²⁺ release, respectively[20]. Therefore, sustained Ca²⁺ signaling through GPCRs requires continued phosphorylation of plasma membrane (PM) phosphatidylinositol (PI) to generate phosphatidylinositol 4-phosphate (PI4P), the precursor of PI(4,5)P₂. Indeed, real-time monitoring of PM phospholipid levels during GPCR signaling reveals that, concomitant with PI(4,5)P₂ depletion, phosphatidylinositol 4-kinase IIIα (PI4KA) is activated to increase PI4P synthesis[21,22].

PI4KA is recruited to the PM by at least two accessory proteins, EFR3A/B and TTC7A/B, which are conserved from yeast to mammals[23–26]. EFR3 is stably associated with the PM via palmitoylation, and anchors the complex to the membrane, while TTC7 (Ypp1 in yeast) binds to both EFR3 and PI4KA (Stt4 in yeast) and acts as the shuttle[27]. A third protein, either FAM126A (Hyccin) or FAM126B, is an essential regulatory component, present only in higher eukaryotes, that stabilizes the TTC7-PI4KA interaction in the cytosol, and enhances PI4KA recruitment to the PM[24]. PI4KA/TTC7/FAM126A heterotrimers form a dimer, and this super-assembly likely stabilizes and orients the PI4KA active site toward the membrane to promote its activity[28]. Furthermore, the disordered C-terminus of FAM126A is not visible in existing structures, and modulates the PI4KA catalytic activity in vitro through an unknown mechanism[29]. This intricate structure suggests that both the assembly and activity of the PI4KA complex are tightly regulated. In yeast, PI4KA recruitment to the PM is regulated by phosphorylation[26]. However, regulation of PI4KA complex assembly and/or activity in mammals remains to be elucidated.

This work discovers CN functions by focusing on the CNAβ1/CNB isozyme. We demonstrate that unlike the cytosolic, canonical CNAβ2, CNAβ1 localizes to cellular membranes, primarily the PM and Golgi apparatus, via palmitoylation of two conserved cysteines within its unique C-terminus. The ABHD17A thioesterase depalmitoylates CNAβ1 causing its redistribution and suggesting that this dynamic palmitoylation regulates CNAβ1 signaling in vivo. To identify potential substrates of CNAβ1, we carried out affinity purification coupled to mass spectrometry (AP-MS) which revealed CNAβ1-specific interactors to be largely membrane-associated, and unexpectedly identified all four members of the PI4KA complex. Using in vivo and in vitro analyses, including hydrogen deuterium exchange mass spectrometry (HDX-MS), we identified multiple sites of CN-PI4KA complex association, including direct interaction with a short linear motif, PSISIT, within the unstructured C-terminal tail of FAM126A. Our studies establish FAM126A as a CN substrate that preferentially interacts with CNAβ1 at the PM. Finally, we uncover a role for CN in PI4P production at the PM by PI4KA during signaling from the type-3 muscarinic receptor. In total, this work discovers a CN-regulated signaling pathway that highlights the PI4KA complex as a regulatory target and demonstrates that palmitoylation dictates substrate specificity of the non-canonical CNAβ1 isoform.

## Results

### CNAβ1 localizes to the plasma membrane, Golgi apparatus, and intracellular vesicles. We sought to investigate the unique functions of CNAβ1 by characterizing it's in vivo properties.

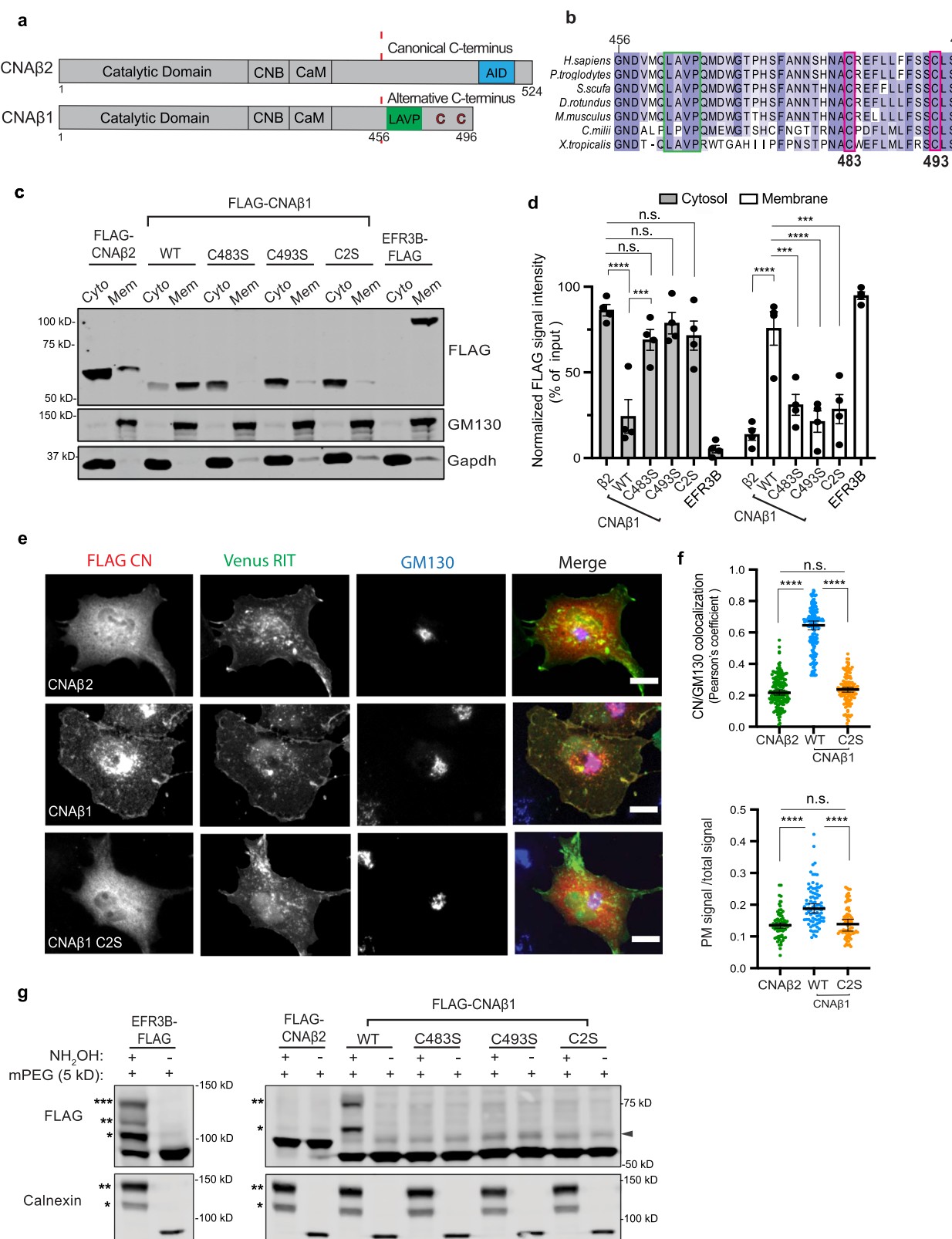

First, we analyzed the intracellular distribution of CNAβ1, which was previously found to be Golgi-associated in mouse embryonic stem cells[19]. Subcellular fractionation of COS-7 cells expressing FLAG-tagged CNAβ1 or CNAβ2, confirmed that CNAβ1 was highly enriched in membranes while CNAβ2 was primarily cytosolic (Fig. 1c, d). Furthermore, indirect immunofluorescence of these cells revealed that CNAβ1 localized to the PM, where it overlapped significantly with a co-expressed PM marker (Venus-RIT)[30], the Golgi apparatus, where it co-localized with GM130, and to intracellular vesicles. By contrast, CNAβ2 was predominantly cytosolic with minimal co-localization with either membrane marker (Fig. 1e, f). Similar distributions were observed in HeLa cells (Supplementary Fig. 1a).

**Fig. 1 CNAβ1 localizes to intracellular membranes via palmitoylation at two conserved cysteines unique to its C-terminal tail. a** Schematic of CNAβ isoforms. CNB and calmodulin binding domains (CNB and CaM); autoinhibitory domain, (AID, blue); LAVP autoinhibitory sequence (green); palmitoylated cysteines (red). **b** CNAβ1 C-terminal (a.a. 456-496) sequence alignment; autoinhibitory LAVP (green) and palmitoylated cysteines (red, C483 and C493), are boxed. **c** Representative immunoblot demonstrating subcellular fractionation of COS-7 cells transfected with FLAG-CNAβ2, -CNAβ1 (WT or cysteine mutants), or EFR3B-FLAG using anti-FLAG. GM130 and Gapdh define membrane and cytosol fractions, respectively ($n = 4$ independent experiments). **d** Quantification of four independent experiments as in **c**. Data show the mean ± SEM. n.s. not significant, ***$p < 0.001$: cytosol, WT vs C483S $p = 0.0004$; membrane, WT vs C483S $p = 0.0004$, WT vs C2S $p = 0.0002$; ****$p < 0.0001$ using two-way ANOVA with Holm-Sidaks multiple comparison tests. **e** Representative images of COS-7 cells expressing FLAG-tagged CNAβ2, CNAβ1 or CNAβ1$^{C2S}$ with Venus-Rit (PM, green). Fixed cells immunostained with anti-FLAG (red) and anti-GM130 (blue). Scale bar = 15 μm. **f** Top graph: Co-localization of FLAG signal (as in **e**) with Golgi marker, GM130. Data show median of Pearson's coefficients with 95% confidence intervals (CI), from ≥100 cells analyzed in three independent experiments (see Statistical Analysis). n.s not significant; ****$p < 0.0001$, using one-way ANOVA followed by Kruskal–Wallis test. Bottom graph: PM localization quantified as anti-FLAG signal intensity at cell periphery (defined in Supplementary Fig. 1d) over total cell intensity (see Methods section). Data show median with 95% CI from ≥70 cells imaged in three independent experiments. ****$p < 0.0001$ using one-way ANOVA followed by Kruskal–Wallis test. **g** Representative immunoblot of Acyl-PEG exchange performed on lysates of COS-7 cells transfected with FLAG-CNAβ2, FLAG -CNAβ1 (WT or cysteine mutants) or EFR3B-FLAG. The number of PEGylation (reflecting S-palmitoylation) events are indicated by asterisks. Arrowhead indicates non-specific antibody band. $n \geq 3$ independent experiments for all constructs (see Statistical Analysis).

**CNAβ1 is palmitoylated at two conserved cysteines unique to its C-terminal tail**. S-Palmitoylation, the reversible addition of a 16-carbon fatty acid chain to cysteine residues via a thioester linkage, allows proteins lacking a transmembrane domain to associate with cellular membranes[31]. We noted that the C-terminus of CNAβ1 contains two highly conserved cysteines: C483, which is contained within a previously defined "Golgi localization domain"[19,32] and C493, which is predicted as a high-confidence S-palmitoylation site[33] (Fig. 1a, b). Thus, we investigated possible palmitoylation of CNAβ1 using acyl resin-assisted capture (Acyl-RAC), during which the thioester linkage in palmitoylated cysteines is cleaved with hydroxylamine ($NH_2OH$) to allow protein binding to thiopropyl-sepharose beads[34]. Acyl-RAC analysis of FLAG-CNAβ2, FLAG-CNAβ1, or EFR3B-FLAG expressed in COS-7 cells revealed the presence of S-palmitoylated cysteines in our positive control, EFR3B and in CNAβ1, but not CNAβ2 (Supplementary Fig. 1b). Furthermore, CNAβ1 mutants containing either single or double serine substitutions at C483 and/or C493, from here on referred to as CNAβ1$^{C483S}$, CNAβ1$^{C493S}$, and CNAβ1$^{C2S}$ respectively, were not captured by Acyl-RAC, suggesting that at least one of these residues is palmitoylated (Supplementary Fig. 1b). To determine the stoichiometry of CNAβ1 palmitoylation, we used acyl-PEG exchange (APE) in which the palmitate groups on modified cysteines are removed by hydroxylamine and replaced with a mass-tag (mPEG) that causes a 5 kDa mass-shift for each acylated cysteine[35]. Mass-tag conversion may be incomplete during APE; thus, this method accurately reports S-acylation states, but not the fraction of protein in the sample that is modified[36]. Our positive controls were EFR3B, which contains three palmitoylated cysteines, and the endogenous ER chaperone, calnexin, which is dually palmitoylated[27,37]. As expected, EFR3B-FLAG and calnexin showed three and two mass-shifted bands, respectively, while cytosolic FLAG-CNAβ2 showed no shifts (Fig. 1g). Interestingly, FLAG-CNAβ1 displayed two mass-shifted forms indicating two sites of palmitoylation, but mutants, CNAβ1$^{C483S}$, CNAβ1$^{C493S}$ or CNAβ1$^{C2S}$, displayed no electrophoretic shifts. Thus, both CNAβ1 cysteines are apparently required for stable palmitoylation suggesting that palmitoylation is cooperative, as described for calnexin[37].

**Palmitoylation is required for CNAβ1 membrane association**. To determine if palmitoylation mediates CNAβ1 membrane association, we first metabolically labeled COS-7 cells with the palmitate analog, 17-octadecynoic acid (17-ODYA) and showed that upon subcellular fractionation, the majority of the 17-ODYA-labeled CNAβ1 was in the membrane fraction

(Supplementary Fig. 1c). Next, we analyzed the fractionation of palmitoylation-defective mutants CNAβ1$^{C483S}$, CNAβ1$^{C493S}$, and CNAβ1$^{C2S}$ which, in contrast to wildtype CNAβ1, were predominantly enriched in the cytosolic fractions (Fig. 1c, d). Finally, we examined each mutant using indirect immunofluorescence. As expected, FLAG-CNAβ1$^{C493S}$ and FLAG-CNAβ1$^{C2S}$ mutants were cytosolic and did not co-localize with either PM or Golgi membrane markers (Venus RIT and GM130, respectively) (Fig. 1e, f and Supplementary Fig. 1e, f). Interestingly, although FLAG-CNAβ1$^{C483S}$ was predominantly cytosolic, a minority of cells exhibited weak Golgi and PM localization (Supplementary Fig. 1e (top panel, red and green boxes), f), suggesting that this mutant might be palmitoylated at a low level that is insufficient for stable membrane association. Thus, palmitoylation of Cys493 may serve as the priming site for Cys483 palmitoylation. In sum, these analyses show that both Cys483 and Cys493 are palmitoylated and that dual palmitoylation is required for the stable association of CNAβ1 with membranes, particularly with the PM. Therefore, the unique lipidated C-terminal tail of CNAβ1 confers distinct localization to this isoform.

**CNAβ1 palmitoylation is dynamically regulated**. Protein palmitoylation is reversed by depalmitoylases, and some proteins, including RAS GTPases, undergo rapid palmitate turnover during their lifespan, which regulates their localization and function[38,39]. To determine palmitoylation dynamics for CNAβ1, we monitored S-acylation while simultaneously controlling for protein turnover using a dual label pulse-chase experiment. Cells were labelled with both palmitate (17-ODYA) and methionine (L-azidohomoalanine or L-AHA) analogs, which were later visualized via click chemistry using AF647-azide and AF488-alkyne, respectively[39–41]. COS-7 cells expressing FLAG-CNAβ1 were briefly labelled with both analogs and then chased with media lacking the analogs but containing either Palmostatin B (Palm B), a pan inhibitor of depalmitoylases, or vehicle (DMSO) (Fig. 2a). FLAG-CNAβ1 was then immunopurified and the level of 17-ODYA incorporated into the pool of L-AHA labelled CNAβ1 (17-ODYA/L-AHA) was determined as a function of time. Palm B treatment caused a marked increase in the ratio of 17-ODYA/L-AHA labelled FLAG-CNAβ1 over time relative to control cells where palmitate turnover occurred (Fig. 2b, c). Thus, palmitoylation of CNAβ1 is dynamic and is actively reversed by depalmitoylases in vivo.

In mammals, two classes of thioesterases are responsible for protein depalmitoylation. The soluble, acyl protein thioesterases (APT1, APT2) and the membrane-associated α/β hydrolase domain proteins (ABHD17s) display distinct substrate

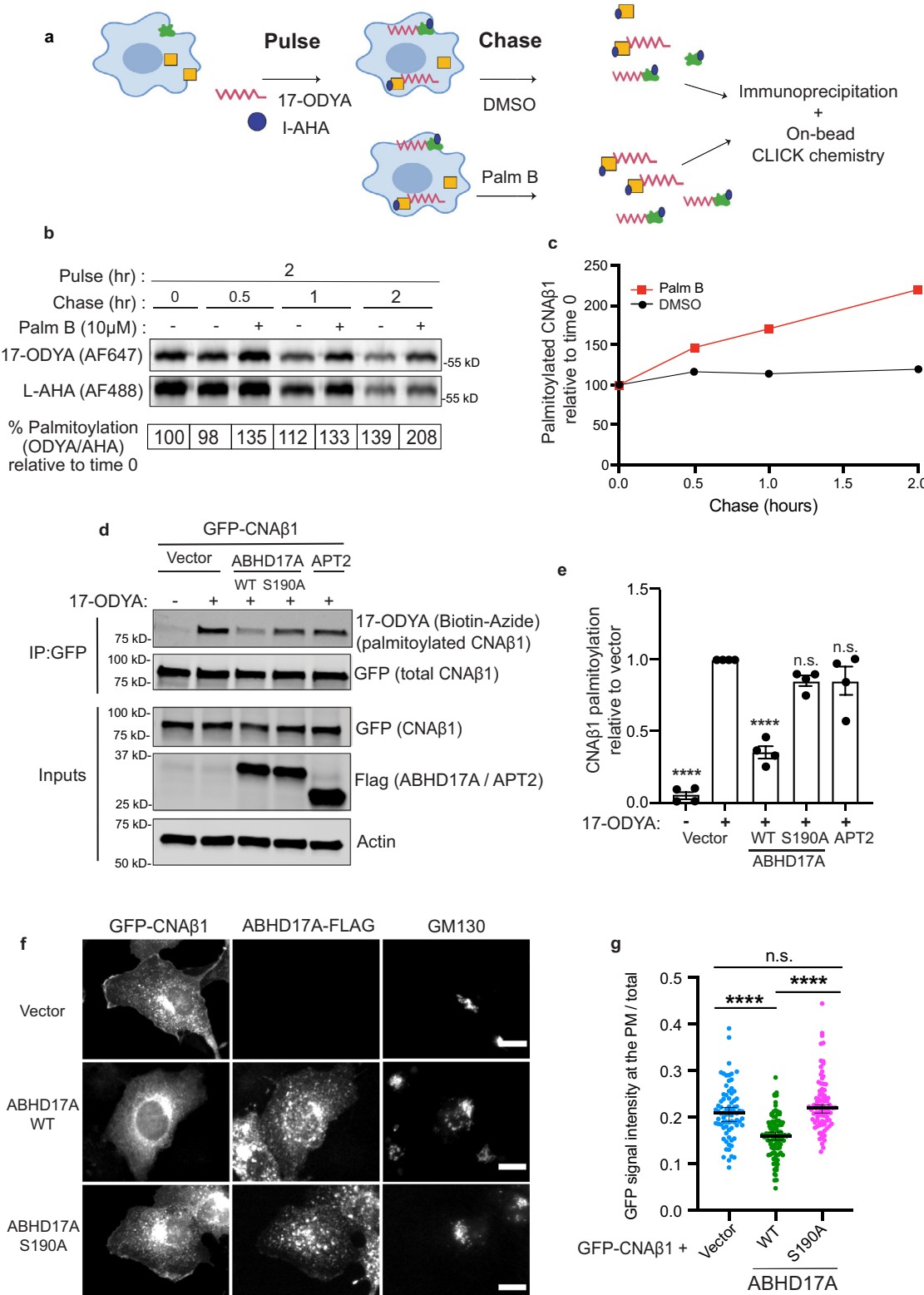

specificities likely due to their differential subcellular distribution[39,40]. Among the ABHD17 family, we focused on the well-characterized ABHD17A, which localizes to the PM, Golgi, and endosomes[39]. To identify which of these thioesterases act on CNAβ1, we examined 17-ODYA labeling of GFP-CNAβ1 co-expressed in COS-7 cells together with a vector, ABHD17A-FLAG (WT or the catalytically impaired mutant S190A[39]),

FLAG-APT2, or mCherry-APT1. Overexpression of ABHD17A WT, but not S190A, dramatically decreased the 17-ODYA labeling of GFP-CNAβ1 (Fig. 2d, e), and resulted in redistribution of GFP-CNAβ1 from the PM to the cytosol and Golgi (Fig. 2f, g), suggesting that ABHD17A depalmitoylates CNAβ1. Furthermore, APE analysis showed that, compared to vector and ABHD17A (S190A), expression of ABHD17A WT significantly reduced

**Fig. 2 CNAβ1 palmitoylation is dynamic: ABHD17A expression promotes CNAβ1 depalmitoylation and alters CNAβ1 subcellular localization. a**
Schematic diagram of the pulse-chase experiment using analogs of palmitate (17-ODYA) and methionine (L-AHA) coupled to CLICK chemistry used in this
study. **b** Pulse-chase analysis of palmitate turnover on FLAG-CNAβ1 by dual-click chemistry as described in **a** in the presence of DMSO or pan-
depalmitoylase inhibitor Palm B. Representative in-gel fluorescence scans showing dual detection of 17-ODYA and L-AHA using Alexa Fluor 647 and Alexa
Fluor 488, respectively. **c** Time course of FLAG-CNAβ1 depalmitoylation in DMSO- and Palm B-treated cells after normalizing 17-ODYA to L-AHA signals
at each chase time. Data shown are mean of each time point from two independent experiments. **d** Analysis of GFP-CNAβ1 palmitoylation co-expressed
with vector, ABHD17A-FLAG (WT or S190A) or FLAG-APT2, using metabolic labeling with 17-ODYA. Representative immunoblot illustrates total CNAβ1
using anti-GFP and 17-ODYA detected using streptavidin following CLICK chemistry with Biotin-Azide. Anti-FLAG shows amount of ABHD17A and APT2
expression ($n = 4$ independent experiments). **e** GFP-CNAβ1 palmitoylation (as in **d**) is quantified by the streptavidin signal (17-ODYA)/total protein signal
(GFP) and normalized to vector control. Data are mean ± SEM. ($n = 4$ independent experiments) n.s. not significant, ****$p < 0.0001$ using one-way ANOVA
with Dunnett's multiple comparison tests. **f** Representative images of fixed, COS-7 cells co-expressing GFP-CNAβ1 with vector, ABHD17A-FLAG (WT or
S190A) immunostained with anti-FLAG and anti-GM130 (Golgi). Scale bar = 15 μm. **g** Images (as in **f**) quantified as GFP signal at the PM relative to total
GFP signal intensity; data show median with 95% confidence intervals. ≥75 cells quantified per condition from four independent experiments. (see
Statistical Analysis). n.s. not significant, ****$p < 0.0001$ using one-way ANOVA followed by Kruskal–Wallis test.

palmitoylation of CNAβ1, especially the dually palmitoylated
form that is required for stable PM association (Supplementary
Fig. 2a). This is consistent with CNAβ1 localization to the Golgi
in these cells. In contrast, overexpression of APT2 (Fig. 2d, e) or
APT1 (Supplementary Fig. 2b, c) did not alter palmitoylation of
CNAβ1. Together, these data reveal that CNAβ1 is dynamically
palmitoylated, which may regulate its localization in vivo. In sum,
we show distinct cellular properties for CNAβ1, compared to
canonical CN isoforms, which led us to search for CNAβ1-
specific substrates and functions.

**Affinity purification and mass spectrometry identifies CNAβ1-
specific interactors.** Previous studies report that, unlike canonical
CN isoforms, CNAβ1 does not activate or interact with NFAT[16].
Indeed, when FLAG-NFATC1 was co-expressed in HEK293 Flp-
In T-REx cell lines that inducibly express GFP, GFP-CNAα, GFP-
CNAβ2, or GFP-CNAβ1, immunoprecipitation of GFP-CNs
confirmed that NFATC1 co-purifies with CNAα and CNAβ2,
but not with CNAβ1. By contrast, the CNB regulatory subunit,
was recovered to the same extent with all three CN isoforms
(Supplementary Fig. 3a). Next, to identify CNAβ1-specific
interactors which might include substrates, we turned to affinity
purification coupled to mass spectrometry (AP-MS). HEK293
Flp-In T-REx cell lines expressing either FLAG-tagged-GFP,
-CNAβtrunc lacking the C-terminal tail (aa 1- 423; truncated
after calmodulin binding site), -CNAβ2, or -CNAβ1 were devel-
oped (Fig. 3a and Supplementary Fig. 3b). This system has been
successfully used to achieve moderate expression levels for sig-
naling proteins and identify biologically relevant interactors for
other protein phosphatases[42–44], although we were unable to
directly compare expression levels of the transgenes with endo-
genous proteins due to the low sensitivity of CNAβ-specific
antibodies. Following immunoprecipitation of each CN, label-
free, quantitative mass spectrometry was used to identify inter-
actors, while comparing with FLAG-tagged-GFP control to
eliminate non-specific binders. In total, 51 high confidence CN-
interacting proteins (defined as those with a bayesian false dis-
covery rate (BFDR) ≤ 1%) were identified (Supplementary Fig. 3c
and Supplementary Data 1). As expected, some established CNA
interactors, including the CNB subunit (PPP3R1) and the inhi-
bitor RCAN3, were identified with all CNAβ constructs (Fig. 3b
and Supplementary Fig. 3c). Of these 51 proteins, 12 were pre-
viously identified CN-interactors and several, including BRUCE,
FAM126A, and GSK3β contain predicted CN-binding motifs
(LxVP or PxIxIT) confirming the validity of our data set[6,7,45].

Excitingly, several proteins preferentially associated with
CNAβ1 relative to CNAβ2 or CNAβtrunc, i.e. spectral counts
≥1.5x more for CNAβ1 than other baits (Fig. 3b), and, consistent
with CNAβ1 localization, were mostly membrane-associated, i.e.

Baculoviral IAP repeat-containing protein 6 (BIRC6/BRUCE),
which localizes to the Golgi and endosomes[46], PM-associated
Phosphorylase B kinase regulatory subunit (PHKB), cell junction
protein Liprin-Beta 1 (PPFIBP1)[47], and endosomal SH3 and BAR
domain-containing protein endophilin B2 (SH3GLB2)[48]. Strik-
ingly, all subunits of the large PM-associated PI4KA complex
were identified: EFR3B, FAM126A (Hyccin), TTC7B, and PI4KA
(PI4KIIIα) (Fig. 3b, c)[23,24,26]. Together, these findings suggest
that CNAβ1 interacts with a unique set of membrane-associated
proteins which may represent CNAβ1-regulated substrates and
pathways.

**CNAβ1 interacts with the PI4KA complex at the plasma
membrane.** We further examined the CNAβ1 interaction with
the PI4KA complex, which is endogenously expressed at very low
levels. Therefore, we engineered a single plasmid that encodes
tagged EFR3B, TTC7B, and FAM126A separated by the viral 2A
linkers, T2A and P2A, respectively, which are cleaved during
translation to ensure balanced expression of each protein with
constant stoichiometry (Supplementary Fig. 3d)[49]. Proper
expression of each component was verified (Supplementary
Fig. 3e) and expected PM localizations for EFR3B, TTC7B,
FAM126A (Supplementary Fig. 3f) and co-expressed PI4KA
(Supplementary Fig. 5c) were confirmed by indirect immuno-
fluorescence. Using this expression system, we first validated
preferential interaction of the PI4KA complex with CNAβ1
compared to CNAβ2 by immunoprecipitation. HEK293 Flp-In T-
REx cells inducibly expressing GFP-FLAG control, GFP-CNAβ2,
or GFP-CNAβ1 were transfected with the EFR3B-HA, TTC7B-
MYC, FLAG-FAM126A-containing plasmid together with GFP-
PI4KA. EFR3B-HA was immunoprecipitated from cell lysates and
co-purifying proteins were analyzed. As expected, GFP-PI4KA
efficiently co-purified with EFR3B indicating functional complex
formation (Fig. 3d, e). Supporting the AP-MS results, significantly
more GFP-CNAβ1 associated with EFR3B compared to GFP-
CNAβ2 or palmitoylation-defective GFP-CNAβ1[C2S]. Thus,
CNAβ1 preferentially interacts with the PI4KA complex due to its
unique PM localization, which is mediated by palmitoylation.

**FAM126A has a putative PxIxIT motif that mediates binding
to CN.** A highly conserved sequence in the intrinsically dis-
ordered C-terminal tail of FAM126A, [512]PSISIT[517], which mat-
ches the consensus of the CN-binding PxIxIT motif was
identified[6,7] (Fig. 4a and Supplementary Fig. 4a). To examine CN
binding, we fused a 16-mer peptide containing this FAM126A
sequence to GST and tested its co-purification with the recom-
binant, HIS-tagged CN heterodimer in vitro, using the PxIxIT
sequence from NFATC1 as a positive control. As expected, the

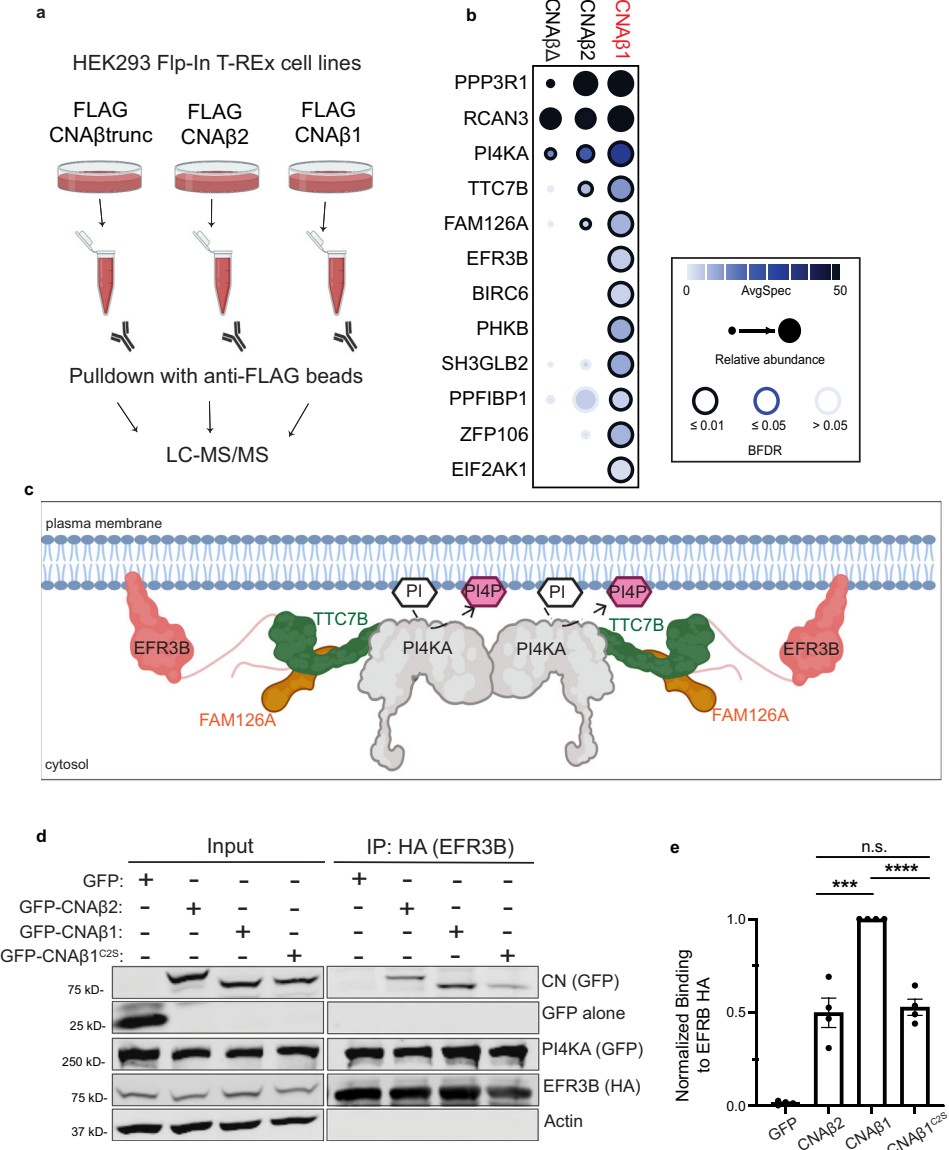

**Fig. 3 CNAβ1-enriched interactors are membrane-associated and include all PI4KA complex members. a** Schematic of experimental plan for AP-MS analyses (created using BioRender.com). **b** Dotplot of AP-MS results including CNAβ1-enriched interactors (those with spectral counts ≥ 1.5x more for CNAβ1 than other baits). Node edge color corresponds to bayesian false discovery rate (BFDR), node size displays prey abundance and node darkness represents number of spectral counts. Full results reported in Supplementary Fig. 3c and Supplementary Data 1. **c** Cartoon representation of the structural organization of the phosphatidylinositol 4-kinase complex containing PI4KA (gray), FAM126A (orange), TTC7B (green) and EFR3B (pink). PI Phosphatidylinositol (white), PI4P phosphatidylinositol 4-phosphate (purple). **d** Immunoblot analysis of anti-GFP immunoprecipitates from inducible Flp-In-T-REx cells expressing GFP-CNAβ2, CNAβ1, or CNAβ1$^{C2S}$, transfected with EFR3B-HA, TTC7B-MYC, FLAG-FAM126A, and GFP-PI4KA. ($n = 4$ independent experiments) **e** Amount of GFP-CNAβ2 and GFP-CNAβ1$^{C2S}$ co-purified with EFR3B-HA, quantified as bound GFP signal/bound HA signal normalized to input. Data are mean ± SEM ($n = 4$ independent experiments). n.s. not significant ($p = 0.7$), ***$p = 0.0007$, ****$p < 0.0001$ using unpaired, two-tailed $t$-test.

FAM126A peptide efficiently co-purified with wild-type HIS-CN but not with mutant CN (NIR) which is defective for PxIxIT-docking[50] (Supplementary Fig. 4b, c). This sequence interacts directly with CN in vitro, as mutating key residues to alanine (FAM126A$^{ASASAA}$), disrupted CN-binding.

Next, to investigate PxIxIT-dependent FAM126A-CNAβ1 association in vivo, we used proximity-dependent labeling (BioID) with the promiscuous biotin ligase, BirA*, which sensitively detects the low affinity interaction of CN with substrates[6,51,52]. We transfected HeLa cells expressing BirA-CNAβ1 with HA-PI4KA and the EFR3B-HA, TTC7B-MYC,

FLAG-FAM126A-containing plasmid described above with either FAM126A$^{WT}$ or CN-binding-defective FAM126A$^{ASASAA}$. Consistent with AP-MS results, each component of the PI4KA complex was biotinylated by BirA-CNAβ1 and as expected, FAM126A$^{WT}$ was significantly more biotinylated than FAM126A$^{ASASAA}$ (Fig. 4b, c). Interestingly, biotinylation of other complex members, i.e, TTC7B, PI4KA, and EFR3B was also reduced in the presence of FAM126A$^{ASASAA}$. In sum, these findings identify PSISIT as a direct a CN-binding PxIxIT motif in FAM126A and suggest that this sequence promotes interaction of CNAβ1 with the entire PI4KA complex.

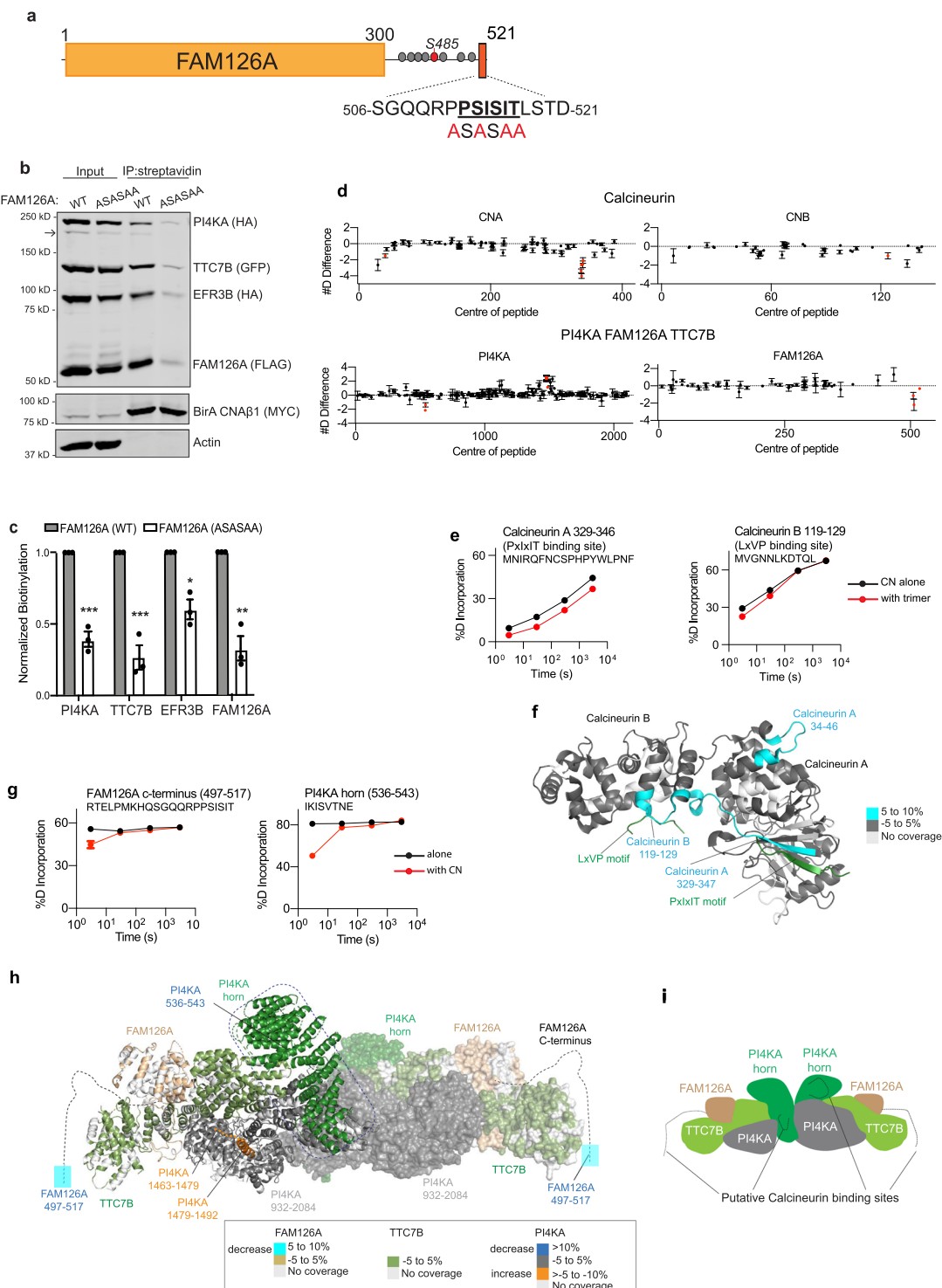

**Hydrogen/deuterium exchange maps CN- PI4KA complex interaction sites**. The cryo-EM structure of PI4KA-TTC7B-FAM126A fails to resolve the PxIxIT-containing, unstructured, disordered C-terminal tail of FAM126A[28]. Therefore, to map the CN-PI4KA complex interaction and identify any conformational changes that occur upon binding, we turned to HDX-MS[29,53]. HDX-MS measures the exchange rate of amide hydrogens with deuterium-containing buffer, which sensitively probes secondary structure dynamics and solvent accessibility[54]. The CNA/CNB heterodimer was produced in *Escherichia coli* and recombinant PI4KA in complex with TTC7B and FAM126A was purified from

insect cells as previously described[29]. The PI4KA/TTC7B/FAM126A trimer and CNA/CNB were exposed to pulses of deuterium when incubated alone or together, with CN in excess over the PI4KA trimer. Localization of differences in HDX requires proteolysis into peptides, with sequence coverage for PI4KA, FAM126A, TTC7B, CNA, and CNB of 77.6%, 80.9%, 84.2%, 89%, and 89.3%, respectively (Supplementary Table 1). Following addition of deuterium-containing buffer ($D_2O$), reactions were quenched at indicated times (3 s, 30 s, 300 s, 3000 s) and the resulting shifts in mass upon deuterium incorporation were analyzed via mass spectrometry. Peptides that showed

**Fig. 4 CN-PI4KA complex interactions include a PxIxIT motif in FAM126A. a** FAM126A schematic showing PxIxIT motif (PSISIT, bold)-containing peptide and mutations (ASASAA, red); phosphorylated residues (gray circles) including Ser 485 (red circle). **b** Representative immunoblot showing proximity-dependent biotinylation analysis of the expressed PI4KA complex containing FLAG-FAM126A (WT or ASASAA) with MYC-BirA*-CNAβ1 in HeLa cells; arrow indicates uncut P2A protein ($n = 3$ independent experiments). **c** Biotinylation of each (as in **b**) quantified as respective bound signal/MYC bound signal normalized to respective signal/Actin signal in inputs. Data show mean ± SEM ($n = 3$ independent experiments). n.s. not significant, *$p = 0.0037$, **$p = 0.0014$, ***$p = 0.000262$ for PI4KA, ***$p = 0.000886$ for TTC7B using multiple unpaired t-tests using Sidak Method. **d, e** HDX data for CN (CNA/CNB) and the PI4KA/TTC7B/FAM126A trimer. ($n = 3$ independent replicates) **d** Sum of the differences in the number of deuterons incorporated for all analyzed peptides over the HDX time course shown for proteins that differ significantly in apo vs. complex state. Peptides with significant HDX (>5%, >0.5 Da, and an unpaired, two-tailed t-test $p < 0.01$) (red); central residue of each peptide is plotted. **e** Deuterium incorporation differences between selected CNA and CNB peptides in the presence (red) or absence (black) of PI4KA/TTC7B/FAM126A trimer are shown. **f** Peptides with maximum significant HDX differences in CNA/CNB upon incubation with PI4KA trimer mapped onto the structure of CN[57] (PDB: 6NUC), coloring explained in legend; PxIxIT and LxVP motifs of the NHE1 peptide (green). **g** Deuterium incorporation in FAM126A and PI4KA peptides displaying significantly decreased amide exchange in the presence (red) vs absence (black) of CN. All error bars in panels **d–g** show the S.D. ($n = 3$ independent replicates), with many being smaller than the size of the point. **h** Maximum significant differences in HDX observed at any timepoint for PI4KA/FAM126A/TTC7B trimer in the presence of CN mapped onto the structure of the PI4KA trimer[28] (PDB: 6BQ1). Dotted lines: unresolved regions in the PI4KA/TTC7/FAM126A structure; colors show differences in exchange as indicated in the legend. **i** Schematic of PI4KA complex showing putative CN-interacting sites. For complete dataset see Supplementary Fig. 4d, e, Supplementary Table 1 and Source Data file.

differences in amide exchange >0.5 Da and >5% at any time point and had unpaired t-test values of $p < 0.01$, across three replicates, were considered significant.

Co-incubation of CNA/CNB with the PI4KA/TTC7B/FAM126A trimer resulted in a large decrease in HDX in the well-characterized PxIxIT-docking groove in CNA[50] (aa 329–346) (Fig. 4d, e, f), consistent with the demonstrated PxIxIT-mediated FAM126A-CN interaction. The N-terminus of CNA (aa 34–36) also displayed decreased amide exchange, suggesting that previously unidentified conformational changes occur upon substrate binding (Fig. 4d, f and Supplementary Fig. 4d, e). Interestingly, a region in CNB that forms part of the LxVP-binding groove[15] (aa 119–129) also showed significantly decreased amide incorporation suggesting additional, as yet unidentified, LxVP-mediated interactions between the PI4KA trimer and CN (Fig. 4d, e, f and Supplementary Fig. 4d, e). As for the PI4KA complex, while no significant changes in amide exchange were seen in TTC7B (Supplementary Fig. 4e), regions in both FAM126A and PI4KA showed significant changes in deuterium exchange in the presence of CNA/CNB. In FAM126A, exchange decreased significantly in the region containing the PSISIT sequence (aa 497–517) consistent with CN-binding to this site (Fig. 4d, g, h, i). In PI4KA an unstructured region within the α-solenoid domain (aka the horn) (aa 536–543) showed a decrease in deuterium incorporation (Fig. 4d, g, h, i), indicating formation of secondary structure either from direct interaction with CN or as an indirect consequence of CN binding to FAM126A. Interestingly, this region contains a PxIxIT-like peptide sequence, "IKISVT", which may be a non-canonical CN-binding motif. In addition, a set of peptides identified in PI4KA between residues 1463–1492 showed increased amide exchange (Fig. 4d, h and Supplementary Fig. 4d, e) revealing a CN-induced conformational change. Overall, these studies indicate multiple sites of contact between CN and PI4KA/TTC7B/FAM126A suggestive of a regulatory interaction.

**FAM126A is a CN substrate.** To examine whether CN regulates phosphorylation of the PI4KA complex we focused on FAM126A because of its small size (~58 kDa), confirmed CN-binding motif and several known phosphorylation sites[55]. First, we expressed FLAG-FAM126A[WT] or CN-binding defective FAM126A[ASASAA], alone or together with TTC7B-MYC and EFR3B-HA and examined their electrophoretic mobility via SDS–PAGE and immunoblot analysis. Slower migrating forms of FAM126A were observed that were enhanced in FAM126A[ASASAA] compared to FAM126A[WT] (labelled PI and PII in Fig. 5a, lane 2 vs 5). Notably,

these shifts were present only when FAM126A was co-expressed with other components (Fig. 5a, lane 1 vs 2 or lane 4 vs 5), especially EFR3B, the membrane anchor for the complex[24] (Supplementary Fig. 5a). These slower migrating forms, indicative of hyperphosphorylation, suggest that FAM126A is phosphorylated only when associated with the PM-localized PI4KA complex i.e. by a PM-associated protein kinase, and that CN dephosphorylates FAM126A in a PxIxIT-dependent manner. To further analyze FAM126A phospho-regulation, we mutated several serine and threonine residues observed to be phosphorylated[55] to the non-phosphorylatable amino acid alanine. Remarkably, mutating serine 485 (FAM126A[S485A]) altered mobility shifts in FAM126A, eliminating PII and reducing PI (Fig. 5a, lane 2 vs 3 and lane 5 vs 6), suggesting that Ser485 is one target of phosphorylation and that additional sites likely contribute to the observed shifts. To analyze the phosphorylation status of Ser485 in FAM126A, we generated a phospho-specific antibody, anti-pFAM126A S485. The specificity of this antibody is demonstrated by analyses of HeLa cells expressing FAM126A mutants (S485A, ASASAA, or ASASAA+S485A) with or without EFR3B and TTC7B co-expression, where this antibody specifically recognized both slower-migrating FAM126A forms (PI and PII, Fig. 5a). Notably, no signal was detected for FAM126A[S485A] or when FAM126A was expressed alone, and the total signal was significantly higher for FAM126A[ASASAA] compared to FAM126A[WT]. Moreover, indirect immunofluorescence using anti-pFAM126A S485 antibody showed enriched signal at the PM, further indicating that FAM126A is phosphorylated when the PI4KA complex is PM-associated (Supplementary Fig. 5b).

Next, we used anti-pFAM126A S485 to probe FAM126A phosphorylation in cells under different signaling conditions. Although direct phosphorylation of the PI4KA complex has not been demonstrated, a recent study identified PKC as a possible regulator of this complex and showed that PMA activates PI4P production at the PM which is blocked by BIM, a PKC inhibitor[22]. Therefore, we monitored FAM126A phosphorylation with anti-pFAM126A S485 under similar conditions by treating cells that co-expressed FLAG-FAM126A (WT or ASASAA mutant), TTC7B and EFR3B upon treatment with combinations of a CN inhibitor (FK506), a PKC activator (PMA) and a PKC inhibitor (BIM). By examining the total intensity of anti-pFAM126 S485 signal (forms PI and PII), we made the following observations (Fig. 5c): First, for cells expressing FAM126A[WT], addition of FK506 significantly increased Ser485 phosphorylation under all conditions (alone or together with PMA, PMA+BIM). Second, compared to FAM126A[WT], cells expressing

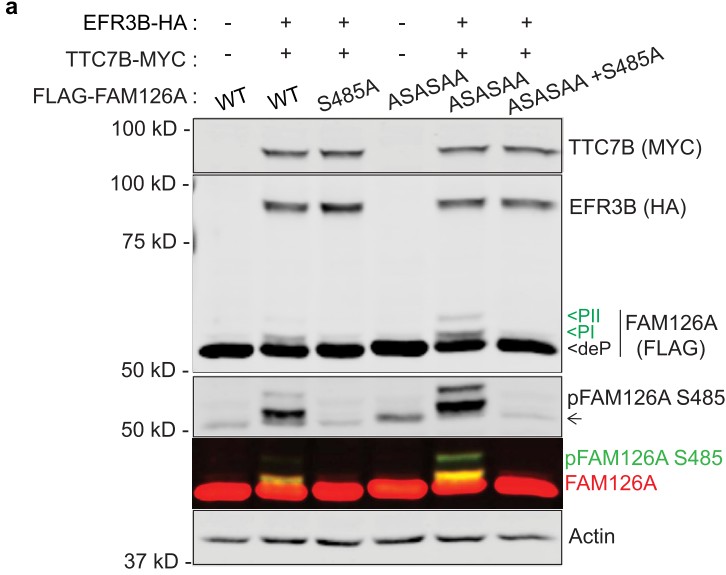

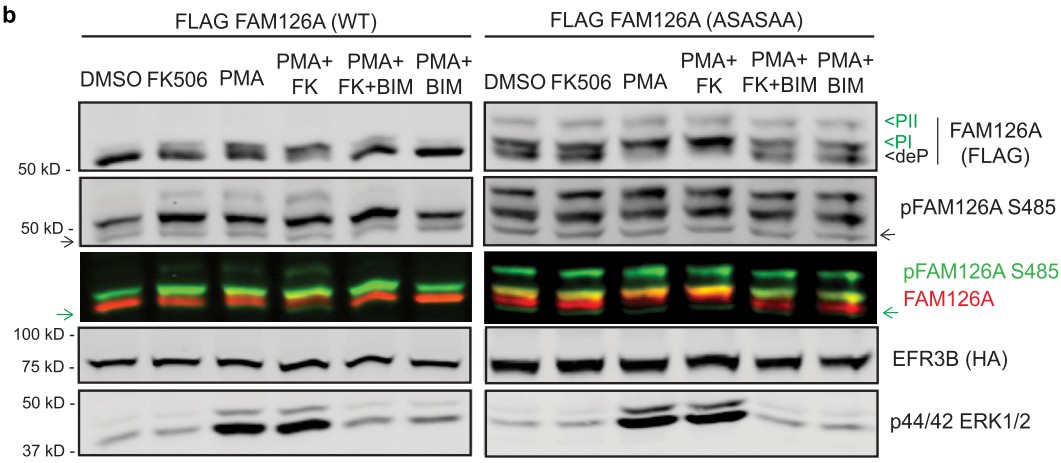

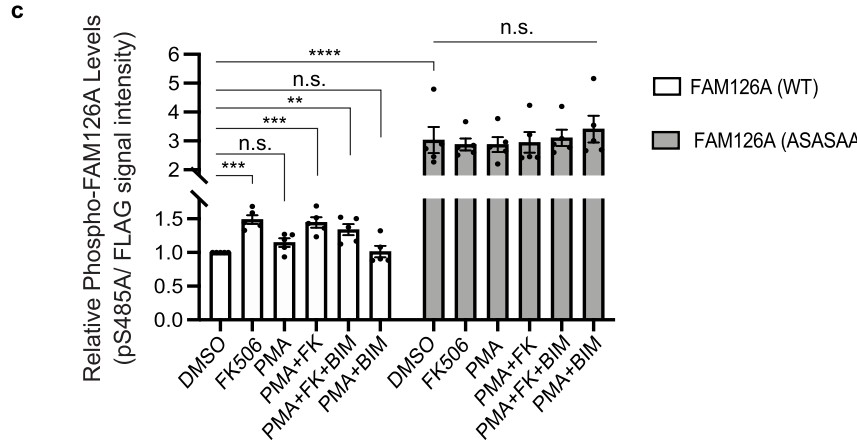

FAM126A$^{ASASAA}$ showed higher levels of Ser485 phosphorylation under all conditions and inhibiting CN with FK506 had no further effect as expected for this CN-binding impaired mutant. Together, these findings show that S485 phosphorylation is CN-regulated. Furthermore, addition of PMA did not enhance S485 phosphorylation which was detected under basal conditions, suggesting that this site is constitutively phosphorylated. Next, we

focused on shifts in electrophoretic mobility of FAM126A (Fig. 5b). Interestingly, for both FAM126A proteins (WT and ASASAA mutant), treatment with PMA caused an electrophoretic shift from the dephosphorylated form (deP) to PI, likely due to phosphorylation of residues other than Ser485. Importantly, this PMA-induced shift was suppressed by BIM indicating PKC dependence. In sum, these findings demonstrate PxIxIT-

**Fig. 5 FAM126A is a CN substrate. a** Representative immunoblot showing electrophoreticmobility shifts observed for FLAG-FAM126A when expressed in HeLa cells. Lysates of cells expressing FLAG-FAM126A (WT, S485A, ASASAA, or ASASAA+S485A) in the presence or absence of EFR3B-HA/ TTC7B-MYC were analyzed using anti-MYC, anti-HA, anti-FLAG (red bands), and phospho-specific pFAM126A S485A (green bands) antibodies. PI and PII phosphorylated states, deP dephoshorylated state ($n = 3$ independent experiments). **b** Representative immunoblot showing analysis of FLAG FAM126A (WT or ASASAA) phosphorylation status in HeLa cells co-expressing FLAG-FAM126A, TTC7B-MYC, and EFR3B-HA across indicated treatments: DMSO (vehicle), FK506 (FK, CN-inhibitor), PMA (PKC activator), BIM (PKC inhibitor) using anti-FLAG (red), anti-HA and anti-pFAM126A S485A (green) antibodies. PKC activation was assessed by phosphorylation of the downstream substrate, ERK using anti-p44/42 ERK 1/2 antibody. Arrows denote non-specific antibody background. ($n = 5$ independent experiments) **c** FAM126A phosphorylation at Ser485 (from **b**) was quantified as the ratio of total pFAM126A S485 signal/ total FLAG signal relative to DMSO-treated FLAG-FAM126A WT signal ratio. Data are mean ± SEM ($n = 5$ independent experiments). **p = 0.0081, ***p < 0.001 ($p = 0.0002$ for DMSO vs FK506; $p = 0.0005$ for DMSO vs PMA + FK), ****p < 0.0001, n.s. (non-significant): $p = 0.53$ for DMSO vs PMA; $p > 0.99$ for the rest, calculated using one-way ANOVAs with Dunnett's multiple comparison tests.

dependent regulation of FAM126A phosphorylation at Ser485 by CN in vivo and establish FAM126A as a CN substrate. These data also reveal PMA-induced phosphorylation of FAM126A, at a distinct site, likely by PM-localized PKC, which might be the molecular basis of the reported regulatory role for PKC in PM PI4P synthesis[22].

**CN regulates PI4P synthesis by the PI4KA complex.** Having shown that CN interacts with the PI4KA complex and that FAM126A is a CN substrate, we next investigated whether CN regulates the assembly and/or activity of this complex. First, we examined interaction of the cytosolic heterotrimer, PI4KA/TTCB/FAM126A, with the membrane anchor EFR3B in the presence of FAM126A$^{WT}$ or CN-binding defective FAM126A$^{A-SASAA}$. Immunopurification of EFR3B-HA showed the same levels of co-purifying GFP-PI4KA, TTC7B-MYC or FLAG-FAM126A with FAM126A$^{WT}$ or FAM126A$^{ASASAA}$ (Fig. 6a, b). Furthermore, indirect immunofluorescence analyses of these cells verified that each component, especially GFP-PI4KA, localized to the PM with either FAM126A$^{WT}$ or FAM126A$^{ASASAA}$, indicating that the complex formed properly (Supplementary Fig. 5c). Thus, no CN-dependent regulation of complex formation via FAM126A was observed.

Next, we explored whether CN regulates PM PI4P synthesis carried out by the PI4KA complex using a previously established bioluminescence resonance energy transfer (BRET) assay that monitors PI4P levels at the PM in live cells during signaling[22]. For this assay, the energy donor (luciferase) is fused to the PI4P binding domain, P4M, of the *Legionella* SidM protein[56], the energy acceptor (Venus) is attached to the PM-targeting sequence from Lck (first ten amino acids, L10) (Fig. 6c), and both proteins are co-expressed with the Gq-coupled muscarinic receptor, M$_3$R, in HEK293T cells. As previously reported, PM PI4P levels transiently increased in control cells (blue lines, Fig. 6d) following addition of the M$_3$R ligand, carbachol ($10^{-7}$ M), due to activation of the PI4KA complex[22]. Excitingly, pre-treatment of these cells with CN inhibitors, FK506 (1 μM) or CsA (10 μM), significantly reduced the level of PI4P produced (red lines, Fig. 6d) consistent with our hypothesis that CN regulates PI4KA complex activity under Ca$^{2+}$ signaling conditions.

In summary, our findings lead us to propose the following model: Signaling from a Gq-coupled GPCR generates an intracellular Ca$^{2+}$ signal that activates CN, and likely PKC, which in turn stimulate the PI4KA complex at the PM to promote PI4P replenishment and thus generating PI(4,5)P$_2$ pools required for sustained signaling (Fig. 7). Our work identifies CNAβ1 as an interaction partner of the PI4KA complex, shows that CN inhibitors alter PI4P production at the PM during signaling, and warrants further investigation into the phosphorylation state of complex components, especially FAM126A and PI4KA, through which CN might be regulating PI4KA activity.

## Discussion

In this study we aimed to discover CN signaling pathways that are regulated by the naturally occurring but understudied CN iso-form, CNAβ1, which is conserved among vertebrates and broadly expressed[10–13]. This isoform differs from canonical CNAβ2 only in its 40 C-terminal residues[10], which confer distinct enzymatic regulation to CNAβ1 through an LxVP-type autoinhibitory sequence (LAVP)[12]. Here we show that the CNAβ1 tail is dually palmitoylated, making CNAβ1 the only known form of CN that directly associates with the PM and Golgi. By contrast, canonical CN isoforms access only select PM proteins that either contain CN-binding sites in their cytosolic domains (e.g. NHE1, TRESK)[57,58] or associate with membrane-anchored scaffolds such as AKAP79[59]. This unique localization determines CNAβ1 substrate specificity including its interaction with all four members of the protein complex that synthesizes the critical phospholipid, PI4P at the PM. We demonstrate that FAM126A, the regulatory component of this complex, is phosphorylated at the PM, directly binds CN, and contains at least one CN-regulated phosphorylation site. These findings led us to discover a hitherto unknown role for CN in regulating PI4P synthesis at the PM during GPCR signaling. The CNAβ1 isoform is ideally positioned to carry out this regulation.

Our finding that CNAβ1 is dynamically palmitoylated has several interesting implications for its regulation in vivo. First, the ability of CNAβ1 to access membrane-associated substrates and hence carry out its functions may be controlled by the palmitoyl transferases (DHHCs) and depalmitoylases that act on it, as has been shown for other signaling enzymes including RAS and LCK[31,39,60,61]. Second, we speculate that palmitoylation-driven binding of the autoinhibitory CNAβ1 tail to membranes may be necessary to fully activate this variant which is only partially activated by Ca$^{2+}$ and CaM in vitro[12]. Thus, examining the enzymes that modify CNAβ1 lipidation will be key for under-standing how CNAβ1 is controlled physiologically. Here we show that a membrane-localized thioesterase ABHD17A, which reg-ulates H- and N-RAS[39], also catalyzes the depalmitoylation of CNAβ1 causing it to redistribute from the PM to the cytosol and the Golgi. Recent work indicates that the ABHD17 family of depalmitoylases specifically targets PM-associated proteins[38], although mechanisms controlling their activity are yet to be identified. Furthermore, determining which of the 23 DHHCs encoded in human genome act on CNAβ1 may provide insights into where and when palmitoylation takes place, as these enzymes exhibit distinct patterns of localization and regulation[31]. In sum, our findings lay the groundwork for further investigation into the role of dynamic palmitoylation in controlling CNAβ1 localization and/or enzymatic activity, which may also provide tools to spe-cifically regulate its functions.

Our investigations identify CN as a regulator of the PI4KA complex composed of PI4KA, TTC7B, FAM126A and EFR3B, and highlight major gaps in our knowledge of how this important

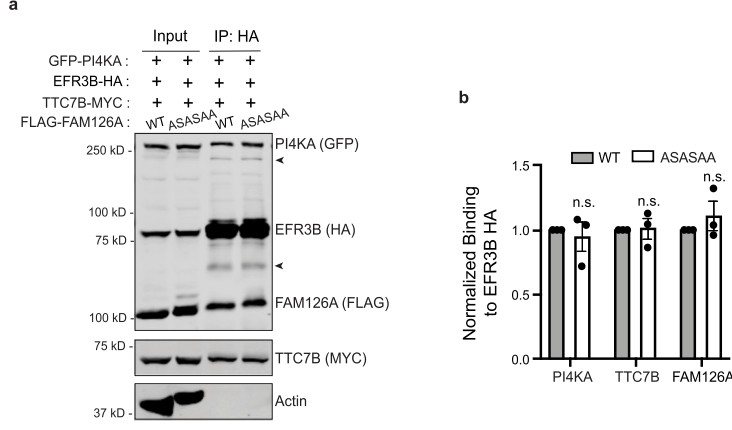

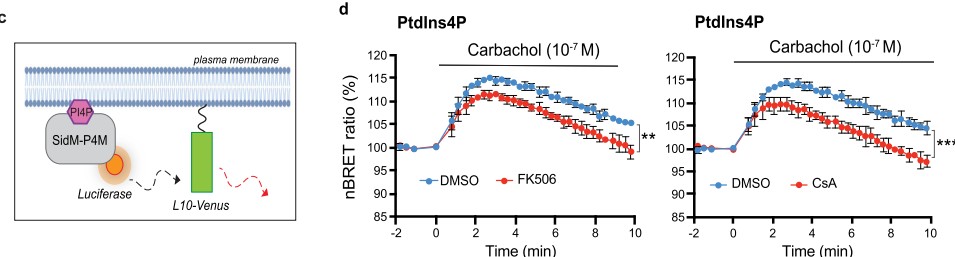

**Fig. 6 CN regulates PI4P synthesis by the PI4KA complex. a** Representative immunoblot showing analysis of the PI4KA complex components in anti-HA immunoprecipitates of HeLa cells expressing GFP-PI4KA, EFR3B-HA/TTC7B-MYC with WT or ASASAA mutant FLAG-FAM126A ($n = 3$ independent experiments). Arrow points to the uncut P2A form. Arrowhead denotes non-specific antibody bands. **b** Co-purification of each component with EFR3B-HA (from **a**) is quantified as respective signals/EFR3B-HA signal in bound fractions normalized to respective signals/Actin signal in input fractions. Data are the mean ± SEM ($n = 3$ independent experiments). n.s. not significant using multiple, unpaired, two-tailed $t$-tests. **c** Cartoon representation of the BRET pair used in experiments shown in **d**. PI4P binding domain of the *Legionella* SidM protein (SidM-P4M, gray) attached to *Renilla* Luciferase (orange) as the donor and Venus (green), targeted to the PM using the first 10 amino acids of Lck, L10, as the acceptor. **d** Normalized BRET ratios reflecting changes in PM PI4P levels in response of carbachol stimulation ($10^{-7}$ M) in HEK293T cells transiently expressing muscarinic receptor, $M_3R$, pre-treated with DMSO (blue), or CN inhibitors (red): FK506 (1 μM) (left) and Cyclosporin A, CsA (10 μM) (right) for 1 hr. Data shown are mean ± SD ($n = 4$ independent experiments). **$p = 0.0063$, ***$p = 0.0004$ using Kolmogorov–Smirnov test.

complex is regulated. Production and maintenance of PM PI4P levels are physiologically critical as evidenced by the wide range of diseases caused by mutations in complex components ranging from neurological (PI4KA), immune and gastrointestinal (TTC7) defects, to hypomyelination and congenital cataracts (FAM126A)[24,28]. Phosphorylation regulates assembly of the PI4KA complex in yeast; however, in mammals, little is known about how the assembly or activity of this complex is modulated. Our interaction studies, including HDX-MS analysis, uncovered potential contacts between CN and multiple PI4KA complex members, and confirmed direct binding to a PxIxIT motif in the C-terminal tail of FAM126A. This tail is completely unstructured and shows no interaction with TTC7B or PI4KA[24], but inhibits PI4KA activity in vitro through an unknown mechanism[29]. Our results provide insights into this regulation by demonstrating that CN binds to and modulates the phosphorylation of at least one site in the FAM126A tail (Ser485) in cells, which, based on our findings, is phosphorylated by an unidentified kinase that is active at the PM under basal conditions. Computational analysis failed to predict any likely candidate kinases for this site (NetPhos 3.1)[62], however up to 10% of human kinases localize to the PM, including many that are uncharacterized[63]. Further studies are required not only to identify relevant kinases, but also to comprehensively map CN-regulated phosphorylation sites in PI4KA complex members and assess the functional consequences of these modifications.

Lastly, our discovery that CN inhibitors reduce PI4P production at the PM induced during $Ca^{2+}$ signaling from Gq-coupled GPCRs suggests that a positive feedback loop exists through which PKC and CN (presumably CNAβ1), regulate the phosphorylation of the PI4KA complex to stimulate its activity and ensure a continued supply of PI4P, the precursor of $PI(4,5)P_2$ (Fig. 7). Evidence for PKC involvement in this stimulation[22] is consistent with the CN-independent, PKC-regulated phosphorylation-shift we observed in FAM126A (Fig. 5b). Rigorously testing this model, however, is challenging due to the complete lack of knowledge about how this large, minimally expressed complex that apparently undergoes extensive allosteric rearrangements[29], is regulated in cells. To date, our attempts to identify changes in PI4P levels or synthesis rates caused by mutations in the FAM126A PxIxIT site or Ser485 have been unsuccessful using overexpression of the proteins. This could be due to limitations in the experimental set up (HEK293T cells overexpressing all PI4KA complex members), or because solely altering FAM126A may not be sufficient to perturb CN-dependent regulation of the complex. Regardless, our work establishes that CNAβ1 preferentially interacts with the PI4KA complex at the PM and suggests that FAM126A is a direct substrate of CNAβ1.

Insights into the physiological functions of CNAβ1 come from studies that overexpress or more recently, delete the CNAβ1 isoform in mice[11,16,18,19]. These knock-out mice are viable, but

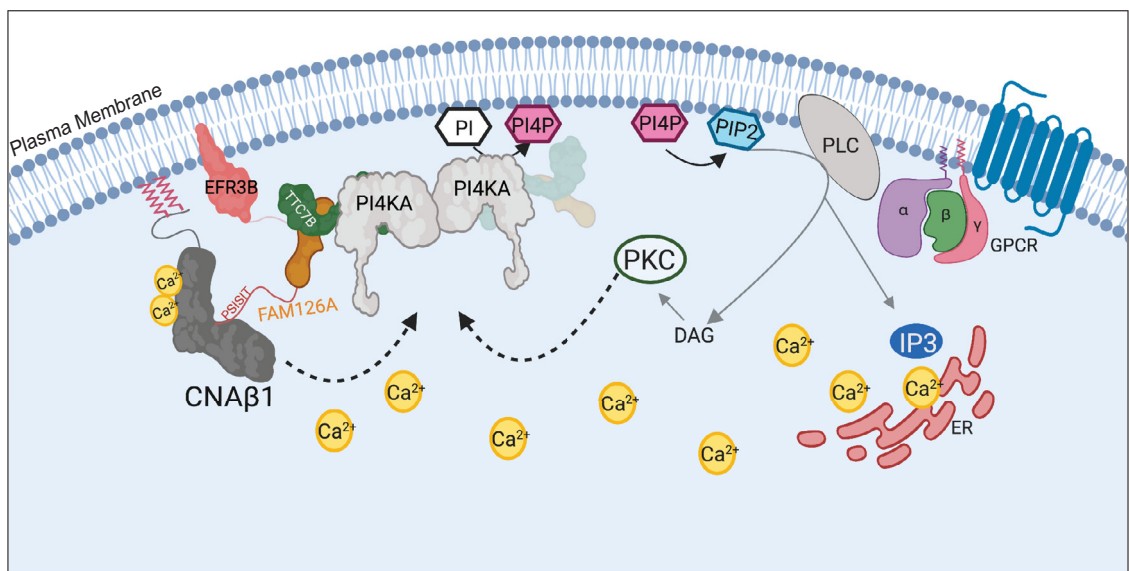

**Fig. 7 Model for CNAβ1 mediated regulation of the PI4KA complex that promotes PI4P synthesis at the PM during GPCR signaling.** Ligand binding and subsequent PLC-mediated cleavage of PI(4,5)P$_2$ generates DAG and IP3, which causes Ca$^{2+}$ release from the ER. This increase in intracellular Ca$^{2+}$ activates CN (CNAβ1), and PKC, which in turn regulate the PI4KA complex at the PM to promote PI4P and PI(4,5)P$_2$ replenishment. See text for details. Schematic generated using BioRender.com.

develop cardiac hypertrophy, possibly due to disruptions in mTORC2/AKT signaling and serine one-carbon metabolism[18]. However, the precise molecular mechanisms underlying these pathologies and whether any of these phenotypes relate to PI4KA complex regulation remain to be determined. Notably, some reports indicate that mTORC2 activity toward AKT takes place at the PM and depends on the PH-domain containing targeting subunit, mSIN1, which binds to phosphoinositides[64–66]. Furthermore, the interactors identified here suggest that CNAβ1 regulates multiple substrates throughout the body. Comprehensive identification of these targets as well as the regulatory mechanisms that control CNAβ1 activity in vivo promise to shed light on Ca$^{2+}$ and CN-regulated pathways and their possible perturbation in patients undergoing long term treatment with CN inhibitors, CsA or FK506/Tacrolimus.

## Methods

**Sequence alignments.** ClustalW was used to create all sequence alignments using Jalview[67]. The following species are used in Fig. 1b: *Homo sapiens* (human, Q5F2F8), *Pan troglodytes* (chimpanzee, A0A2J8NUG2), *Sus scrofa* (pig, A0A480QFW6), *Desmodus rotundus* (Bat, K9ISS2), *Mus musculus* (mouse, Q3UXV4), *Callorhinchus milii* (ghost shark, V9KGC1), and *Xenopus tropicalis* (western clawed frog, A0A6I8R6A9). The following species are used for Supplementary Fig. 4a: *Homo sapiens* (human, Q9BYI3), *Gorilla gorilla* (gorilla, A0A2I2YR80), *Macaca mulatta* (monkey, H9ZEG3), *Sus scrofa* (pig, I3LJX1), *Felis catus* (cat, M3WEC3), *Bos taurus* (bovine, E1BFZ6), *Mus musculus* (mouse, Q6P9N1), *Gallus gallus* (chicken, Q5ZM13), *Xenopus tropicalis* (western clawed frog, F7EHL4), *Callorhinchus milii* (ghost shark, A0A4W3JDV9), and *Danio rerio* (zebrafish, Q6P121).

**Cell culture and transfection.** HeLa, COS-7, and HEK 293T cells were grown at 37 °C in a 5% CO$_2$ atmosphere in cell culture medium (Dulbecco's modified Eagle's medium, DMEM) (CA 10-013, Sigma-Aldrich) supplemented with 10% fetal bovine serum (FBS, Benchmark™ Gemini Bio Products). Cells were transfected as indicated in each experiment using jetOPTIMUS (VWR) as per the manufacturer's instructions. HeLa cells were gifts from the Skotheim lab. COS-7 cells were purchased from ATCC (CRL-1651). Cells were validated via STR profiling[68,69] (see Source Data file).

**Stable cell line generation.** Human [taxid:9606] cells [Flp-In T-REx 293 cells], obtained from Gingras Lab, were transfected in a six-well format with 0.2 μg of tagged DNA in pcDNA5 FRT TO vector and 2 μg pOG44 (OpenFreezer V4134), using lipofectamine 2000 (Life Technologies), according to the manufacturer's

instructions. On day 2, cells were trypsinized, and seeded into 10 cm plates. On day 3, the medium is replaced with DMEM containing 5% fetal bovine serum, 5% calf serum, 100 units/ml penicillin/streptomycin, 3 μg/ml blasticidin (RPI), and 200 μg/ml hygromycin (Sigma-Aldrich). Medium was replaced every 2–4 days until non-transfected cells died and isolated clones were ~1–2 mm in diameter (13–15 days). Pools of cells were generated by trypsinization of the entire plate and replating in fresh selection medium. Pools were amplified to one 15 cm plate. From this plate, cells were trypsinized (volume = 8 ml) and replated in five 15 cm plates. A frozen stock was generated from the plate when cells reached ~80% confluence.

**Plasmids.** DNAs encoding the human CNAβ1(1-496), CNAβ2(1-524) were subcloned into a pcDNA5 expression vector which encodes an N-terminal FLAG tag or GFP tag. FAM126A cDNA received from Addgene, was subcloned into pcDNA5 with N-terminal FLAG tag, between BamHI and XhoI sites. Variants of CNAβ1 (CNAβ1$^{C483S}$, CNAβ1$^{C493S}$ CNAβ1$^{C2S}$) and FAM126A (FAM126A$^{ASASAA}$, FAM126A$^{S485A}$, FAM126A$^{ASASAA+S485A}$) were generated using the Quickchange (Agilent) site-directed mutagenesis kit. Plasmids containing the DNA encoding ABHD17A (WT and S190A mutant), APT1, APT2, and Venus-RIT were gifts from the Conibear lab. CNA/CNB plasmid (residues 2-391 of human CNA alpha isoform and human CNB isoform 1 tandemly fused in pGEX6P3 (which encodes N-terminal GST tag) for protein purification and use in HDX-MS experiments were cloned as described before[6]. 6x His-CNA (residues 1–391 of human CNA alpha isoform and human CNB isoform 1 tandemly fused in, p11 vector) used in in vitro peptide binding assays, was cloned as described before[15]. His-CN NIR ($^{330}$NIR$^{332}$-AAA mutations) generated using site-directed mutagenesis using His-CN WT as template. Plasmids encoding human PI4IIIα were gifts of the Balla lab. Plasmids encoding human EFR3B and TTC7B, both C-terminally tagged, were gifts of the De Camilli lab. EFR3BHA_T2A_TTCBMYC (or GFP)_P2A_FLAG FAM126A plasmid was generated in between HindIII and NotI sites of pcDNA3.1 vector. The DNA sequence that encodes viral T2A (GSGEGRGSLLTCGDVEENPGP) was subcloned to the 5′ end of the EFR3B-HA sequence, TTC7B-MYC was then subcloned in frame to the T2A sequence. FLAG-FAM126A with DNA sequence encoding for P2A (GSGATNFSLLKQAGDVEENPGP) was cloned in frame, at the 5′ end of the TTC7B-MYC sequence. Primers used to generate plasmids can be found in Supplementary Table 2.

**Antibodies.** Commercial antibodies used, along with their working dilutions, are indicated in the methods section for each experiment. The phosphospecific antibody against Serine 485 site in FAM126A was manufactured by 21st Century Biochemicals as follows: a peptide corresponding to the sequence Hydrazine-Ahx-ANRFSAC[pS]LQEEKLI-amide was manufactured by Fmoc chemistry, HPLC purified to >90%, and its mass and sequence were verified by nanospray MS and CID MS/MS, respectively. The peptides, along with carrier proteins and adjuvant, were injected into New Zealand White rabbits using an initial CFA injection, followed by IFA injections. A production bleed was then taken from each of the rabbits. Sera were passed multiple times over a hydrazine reactive resin which was linked to the immunogen peptides, then rinsed with both salt and phosphate

buffers. The antibody fractions were collected using an acidic elution buffer and immediately neutralized before a two-stage dialysis into PBS buffer, pH 7.2. The antibody concentration was determined using a spectrophotometer (A280). The purified antibodies were then passed multiple times over a hydrazine reactive resin, which is linked with the unmodified peptides (those not injected). These immunodepletion steps were done to remove any non-specific/phospho-independent antibodies. The final antibodies were then buffered in a PBS/50% glycerol buffer, pH 7.2 and the final concentration was calculated using a spectrophotometer (A280).

**Immunofluorescence, microscopy, and image analysis**. HeLa or COS-7 cells were grown on 12 mm, #1.5H glass coverslips (ThorLabs). In all, 24 h post-transfection, cells were washed with 1X PBS and fixed in 4% paraformaldehyde (PFA) solution (diluted from 16% PFA, Electron Microscopy Sciences) in PBS for 15 min. Cells were washed thrice with PBS and permeabilized for 5 min in block buffer (1x PBS with 0.2 M Glycine, 2.5% FBS) with 0.1% Triton X-100. Cells were then incubated in block buffer without detergent for 30 min. Coverslips were incubated with primary antibodies diluted in block buffer (without detergent) for 1 h, washed multiple times with 1x PBS followed by incubation with secondary antibodies for 1 h at room temperature. Coverslips were washed again and mounted using Prolong Diamond Antifade mountant (Thermo Fisher). Images were acquired on a single z-plane on Lionheart™ FX automated widefield microscope with a 20X Plan Fluorite WD 6.6 NP 0.45 objective. For Fig. 1e and Supplementary Fig. 1e, primary antibodies used: mouse anti-FLAG, M2 (1:500, Sigma-Aldrich, F1804) and rabbit anti-GM130, D6B1 (1:400, Cell Signaling Technologies, 12480). Secondary antibodies used: anti-mouse Alexa Fluor 647 (1:500, Invitrogen) and anti-rabbit Brilliant Violet 421 (1:100, Biolegend). YFP (500 nm), Texas Red (590 nm), and DAPI (350 nm) filter cubes were used to image Venus, FLAG, and GM130 respectively. For Fig. 2f: GFP (465 nm), Texas Red and DAPI filter cubes were used to image GFP, FLAG, and GM130, respectively.

Image analysis: Image analyses were performed in using FIJI[70] and the EzColocalization[71] plugin was used to determine the Pearson correlation coefficient for co-localization analyses in Fig. 1f and Supplementary Fig. 1f. For PM localization, a binary mask was generated from the thresholded Venus-RIT channel and saved as a selection (outer) to measure total signal intensity cell. The second mask was produced by five iterations of erosion function and subtracted from the outer mask using image calculator. The resulting mask (Supplementary Fig. 1d) was converted to a selection and used to measure the PM signal intensity.

**Detergent-assisted subcellular fractionation**. COS-7 cells were seeded onto 60 mm plates and transfected with FLAG-CNAβ2, FLAG-CNAβ1 (WT or C483S, C493S, C483/C493S) or EFR3B-FLAG at 80% confluency. In all, 48 h post transfection, cells were rinsed, harvested, and pellets snap-frozen in liquid nitrogen. Pellets were resuspended in 200 μl digitonin buffer (10 mM HEPES pH 6.8, 100 mM NaCl, 300 mM sucrose, 3 mM MgCl₂, 5 mM EDTA and 0.015% Digitonin) supplemented with protease inhibitors by pipetting and rotating at 4 °C for 15 min. Input (6%) was taken as input prior to centrifugation at 2000×g for 20 min. The supernatant was removed, centrifuged at 16,000×g for 5 min to remove any contamination from the pellet fraction, and saved as the cytosol fraction. The pellet was washed twice with ice-cold PBS and resuspended in 200 μl Triton X-100 buffer (HEPES pH 7.5, 100 mM NaCl, 300 mM sucrose, 3 mM MgCl₂, 3 mM EDTA, 1% Triton X-100) supplemented with protease inhibitors. Pellets were lysed for 30 min by rotating at 4 °C followed by centrifugation at 7000×g for 10 min and supernatant collected as the membrane fraction. The clarified supernatant is saved as the cytosol fraction. Inputs and equal volumes (6%) of the cytosol and membrane fractions were mixed with 6X SDS sample buffer, heated to 95 °C for 5 min and resolved by SDS–PAGE followed by immunoblotting. Primary antibodies used: anti-FLAG (1:2500; Sigma F3165), rabbit anti-calnexin (1:3000 ADI-SPA-865, Enzo Sciences), and anti-Gapdh (1:20,000, 1E6D9, Proteintech). After incubation with secondary antibodies: IRDye 680RD Goat anti-mouse IgG (H + L) (1:15,000, Li-COR Biosciences 926-68071) and IRDye 800CW Goat anti-rabbit IgG (H + L) (1:15,000, Li-COR Biosciences 926-32211), blots were imaged with the Li-Cor Odyssey imaging system. Enrichment in cytosol fraction was quantified as FLAG signal/Gapdh signal in cytosol fraction normalized to FLAG signal/Gapdh signal in inputs. Similarly for membrane enrichment, FLAG signal/Calnexin signal in membrane fraction normalized to FLAG signal/Calnexin signal in input. Statistical analysis was performed using GraphPad. Uncropped and unprocessed scans of the blots are provided in the Source Data file.

**Acyl-resin assisted capture**. The Acyl-RAC protocol was performed as described previously[34] with minor changes. In brief, COS-7 cells were seeded on 60 mm plates and transfected at 70% confluency with FLAG-CNAβ2, FLAG-CNAβ1 (WT or C483S, C493S, C483/C493S) or EFR3B-FLAG using JetOptimus. In all, 48 h following transfection, cells were harvested in ice-cold PBS and snap frozen in liquid nitrogen. Pellets were lysed in TAE lysis buffer (50 mM TEA pH 7.3, 150 mM NaCl, 2.5% SDS) supplemented with 1 mM PMSF and protease inhibitors, vortexed briefly and incubated at 37 °C for 20 min with constant gentle agitation. Lysates were subjected to fine needle aspiration with sterile 27.5-gauge needle and clarified by centrifugation (16,000×g for 20 min). In all, 400 μg of each lysate was

diluted to 2 mg/ml with lysis buffer and incubated with 10 mM TCEP (646547, Sigma-Aldrich) for 30 min, nutating at room temperature. In total, 25 mM NEM (N-ethylmaleimide, 40526, Alfa Aesar) was then added to the mix and incubated by gentle mixing at 40 °C for 2 h to block free thiols. NEM was removed by acetone precipitation by adding four volumes of ice-cold acetone. Proteins were allowed to precipitate at −20 °C overnight. Following centrifugation of the solution at 16,000×g for 15 min, the pellets were extensively washed with 70% acetone and the pellets were airdried for 5 min at room temperature. Pellets were resuspended in 200 μl of binding buffer (50 mM TEA pH 7.3, 150 mM NaCl, 1 mM EDTA, 1% SDS. 0.2% Triton X-100) by heating at 40 °C with frequent mixing. Approximately 20 μl from each sample was taken as input and the rest were split into two 1.5-ml microcentrifuge tubes. To capture S-palmitoylated proteins, 40 μl prewashed thiopropyl Sepharose 6b (T8387, Sigma-Aldrich, prepared fresh) was added to samples in the presence of either 0.75 M NH₂OH (from 2.5 M stock, pH 7.5, freshly diluted from Hydroxylamine solution (467804, Sigma-Aldrich)) or binding buffer (without SDS and EDTA for the negative control). Binding reactions were carried out on a rotator at 30 °C for 4 h. Resins were washed 4-5x with binding buffer, 5 min each, and proteins were eluted in 30 μl binding buffer supplemented with 50 mM DTT shaking at 30 °C for 30 min. In all, 6x SDS sample buffer was added to the samples followed by heating to 95 °C for 5 min. Inputs and eluates were separated by SDS–PAGE and transferred to nitrocellulose for western blotting with mouse anti- FLAG (1:2500; Sigma F3165) and rabbit anti-calnexin (1:3000 ADI-SPA-865, Enzo Life Sciences) antibodies. After incubation with secondary Li-Cor antibodies, blots were imaged with the Li-Cor Odyssey imaging system. Uncropped and unprocessed scans of the blots are provided in the Source Data file.

**Acyl-PEG exchange**. The previously published protocol[35] was modified as follows: COS-7 cells expressing the indicated proteins were lysed and subjected to reductive alkylation with TCEP and NEM as described in the Acyl-RAC protocol. Following alkylation of total lysate (300–400 μg) proteins were precipitated with four volumes of ice-cold Acetone at −20 °C overnight. Pellets were washed extensively with 70% Acetone, air dried for 5 min and resuspended in 72 μl TEA buffer pH 7.3, with 4% SDS (50 mM TEA, 150 mM NaCl, 0.2% Triton X-100, 4 mM EDTA) by heating to 40 °C for an hour with constant mixing. Lysate was clarified by centrifugation at 16,000×g for 5 min. Approximately 7 μl (10%) from each sample was removed as input, the rest was split into two 30 μl aliquots. For NH₂OH treated sample, 36 μl NH₂OH (2.5 M stock) was added and brought up to 120 μl with TEA buffer with 0.2% Triton X-100 (50 mM TEA, 150 mM NaCl). For negative control not treated with NH₂OH, 90 μl TEA buffer with 0.2% Triton X-100 was added. After incubation at 30 °C for 1 h on a rotator, proteins were precipitated using methanol-chloroform-H₂O, briefly air dried and resuspended in 30 μl TEA buffer with 4% SDS, 50 mM TEA, 150 mM NACl, 0.2% Triton X-100, 4 mM EDTA by gentle mixing at 40 °C. Each sample was treated with 90 μl TEA buffer with 1.33 mM mPEG-Mal (Methoxypolyethylene glycol maleimide, 5 kDa, 63187 Sigma-Aldrich) for a final concentration of 1 mM mPEG-Mal. Samples were incubated for 2 h at RT with agitation before a final methanol-chloroform-H₂O precipitation. The pellets were resuspended in 50 μl TAE lysis buffer (50 mM TEA pH 7.3, 150 mM NaCl, 2.5% SDS) and 10 μl 6X SDS sample buffer was added before heating the sampled for 5 min at 95 °C. Typically 14 μl of each sample was separated by SDS–PAGE and analyzed by immunoblot with FLAG and Calnexin antibodies. After incubation with secondary Li-Cor antibodies, blots were imaged with the Li-Cor Odyssey imaging system. Uncropped and unprocessed scans of the blots are provided in the Source Data file.

**Pulse-chase metabolic labeling with 17-ODYA and L-AHA**. COS-7 cells were transfected with cDNA encoding FLAG-CNAβ1 using Lipofectamine 2000 as per manufacturer's instructions. Twenty hours following transfection, cells were washed in phosphate-buffered saline (PBS) and starved in methionine-free DMEM containing 5% charcoal-filtered FBS (Life Technologies), supplemented with 1 mM L-glutamine and 1 mM sodium pyruvate for 1 h. Cells were then briefly washed in PBS then labeled with 30 μM 17-ODYA and 50 μM L-AHA for 2 h in this media. Labeling media was removed, cells were washed twice in PBS before chasing in complete DMEM supplemented with 10% FBS and 300 μM palmitic acid. Palmostatin B (Palm B) or DMSO (vehicle) were added at chase time 0 and Palm B was replaced every hour. At indicated time points, cells were washed twice in PBS and frozen at −80 °C until processing. Cells were lysed with 500 μl triethanolamine (TEA) lysis buffer (1% Triton X-100, 150 mM NaCl, 50 mM TEA pH 7.4, 100xEDTA-free Halt Protease Inhibitor [Life Technologies]). The lysates were transferred to 1.5-ml Eppendorf tubes (Corning), vigorously shaken while placed on ice in between each agitation. Lysates were cleared by centrifugation at 13,000×g for 15 min at 4 °C. Solubilized proteins in the supernatant were quantified using Bicinchoninic acid (BCA) assay (Life Technologies), 650 μg–1 mg of the lysate was added to Protein A-Sepharose beads (GE Healthcare) pre-incubated for 3–7 h with rabbit anti-FLAG antibody (Sigma-Aldrich) at 4 °C. Immunoprecipitations were carried out overnight rotating at 4 °C.

*Sequential on-bead CuAAC/click chemistry*. Sequential on-bead click chemistry of immunoprecipitated 17-ODYA/L-AHA-labeled proteins was carried out as previously described[39] with minor modifications. After immunoprecipitation, Sepharose beads were washed thrice in RIPA buffer, and on-bead conjugation of

AF647 to 17-ODYA was carried out for 1 h at room temperature in 50 μl of freshly mixed click chemistry reaction mixture containing 1 mM TCEP, 1 mM CuSO₄.5H₂O, 100 μM TBTA, and 100 mM AF647-azide in PBS. After three washes in 500 μl ice-cold RIPA buffer, conjugation of AF488 to L-AHA was carried out for 1 h at room temperature in 50 μl click-chemistry reaction mixture containing 1 mM TCEP, 1 mM CuSO4.5H2O, 100 μM TBTA, and 100 mM AF488-alkyne in RIPA buffer. Beads were washed thrice with RIPA buffer and resuspended in 10 μl SDS buffer (150 mM NaCl, 4% SDS, 50 mM TEA pH 7.4), 4.35 μl 4X SDS-sample buffer (8% SDS, 4% Bromophenol Blue, 200 mM Tris-HCl pH 6.8, 40% Glycerol), and 0.65 μl 2-mercaptoethanol. Samples were heated for 5 min at 90 °C and separated on 10% tris-glycine SDS–PAGE gels for subsequent in-gel fluorescence analyses. A Typhoon Trio scanner (GE Healthcare) was used to measure in-gel fluorescence of SDS–PAGE gels: AF488 signals were acquired using the blue laser (excitation 488 nm) with a 520BP40 emission filter, AF647 signals were acquired using the red laser (excitation 633 nm) with a 670BP30 emission filter. Signals were acquired in the linear range and quantified using the ImageQuant TL7.0 software (GE Healthcare). For pulse-chase analyses, the ratio of palmitoylated substrates : total newly synthesized substrates were calculated as AF647/AF488 values at each time point, normalized to the value at $T = 0$. Uncropped and unprocessed scans of the blots are provided in the Source Data file.

**Determination of palmitate incorporation in the presence of thioesterases.**
COS-7 cells were seeded onto 60 mm plates and transfected with GFP- CNAβ1 together with vector, ABHD17A-FLAG (WT or S190A mutant), FLAG-APT2 or mCherry-APT1. In all, 24 h post-transfection, media was replaced with DMEM containing 2% FBS and labelled with 30 μM 17-ODYA (17-Octadecynoic Acid,34450, Cayman Chemicals) or DMSO for 3 h at 37 °C incubator. Cells for rinsed thrice with ice-cold PBS, harvested, and pellets were snap-frozen in liquid nitrogen. Pellets were then lysed in TEA lysis buffer (50 mM TEA pH 7.4, 150 mM NaCl, 1% Triton X, 1 mM PMSF) supplemented with protease inhibitors by rotating for 20 min at 4 °C. Lysates were subjected to fine-needle aspiration with a sterile 27 G syringe and clarified by centrifugation at 16,000×g for 15 min. In total, 300–400 μg of each lysate was adjusted to 1 mg/ml with TAE lysis buffer and bound to 10 μl pre-washed GFP-trap magnetic particles in for 1–2 h rotating end-over-end at 4 °C. Input (5%) was taken prior to bead binding. Beads were washed thrice in modified RIPA buffer (50 mM TAE pH 7.4, 150 mM NaCl, 1% Triton X-100, 1% sodium deoxycholate, 0.1% SDS). Proteins bound to beads were conjugated to azide-biotin in 50 μl PBS with click chemistry reactants for 1 h at RT with constant agitation. Click chemistry reactants were freshly prepared as a 5X master mix that consists of 0.5 M biotin-azide (Biotin-Picolyl azide, 1167, Click Chemistry Tools), 5 mM TCEP, 0.5 mM TBTA (Tris[(1-benzyl-1H-1,2,3-Triazol-4-yl)methyl]amine, Sigma-Aldrich), and 5 mM CuSO₄.5H₂O. Beads were washed thrice in modified RIPA buffer and proteins eluted by boiling in 2X SDS sample buffer before resolving with SDS–PAGE. Anti-GFP (1:4,000, Living Colors, 632380, Clontech) was used to probe for GFP-CNAβ1, biotin incorporation was detected using fluorophore conjugated Streptavidin antibody (Licor IRDye 800CW Steptavidin, LI-COR Biosciences). ABHD17A and APT2 levels were determined using anti-FLAG (1 : 2,500; F3165, Sigma- Aldrich) and APT1 was detected using for anti-RFP (1 : 3000; 22904, Rockland Inc.). Level of GFP-CNAβ1 palmitoylation was quantified as streptavidin signal normalized to bound GFP signal. Statistical analyses were performed in GraphPad. Uncropped and unprocessed scans of the blots are provided in the Source Data file.

**Affinity purification coupled to mass spectrometry analyses**
*Protein expression and FLAG Affinity purification.* Flp-In T-REx 293 cells at ~60–70% confluence were induced with 1 μg/ml tetracycline for 24 h. Subconfluent cells (~85–95% confluent) were harvested as follows: medium was drained from the plate, 0.5 ml ice-cold PBS was added, and the cells were scraped (using a silicon cake spatula) and transferred to a 1.5 ml Eppendorf tube on ice. Cells were collected by centrifugation (5 min, 1500×g, 4 °C), the PBS aspirated, and cells resuspended in 1 ml ice-cold PBS prior to centrifugation (5 min, 1500×g, 4 °C). This step was repeated once more, the remaining PBS was aspirated, and the weight of the cell pellet was determined. Cell pellets were frozen on dry ice and transferred to −80 °C until processing.

*Affinity purification.* Cells were lysed by passive lysis assisted by freeze-thaw. Briefly, to the frozen cell pellet, a 1:4 pellet weight:volume ratio of ice-cold lysis buffer was added, and the frozen pellet was resuspended by pipetting up and down. The lysis buffer was 50 mM HEPES-NaOH pH 8.0, 100 mM KCl, 2 mM EDTA, 0.1% NP40, 10% glycerol, 1 mM PMSF, 1 mM DTT and Sigma protease inhibitor cocktail, P8340, 1:500. Tubes were frozen and thawed once by placing on dry ice for 5–10 min, then incubated in a 37 °C water bath with agitation until only a small amount of ice remained. Thawed samples were then put on ice, and the lysate transferred to 2 ml Eppendorf tubes. An aliquot (20 μl) was taken to monitor solubility. This aliquot was spun down, the supernatant transferred to a fresh tube, and 6 μl 4X Laemmli sample buffer added. (The pellet was resuspended in 26 μl 2X Laemmli sample buffer). The 2 ml tubes were centrifuged at 14,000 rpm for 20 min at 4 °C, and the supernatant transferred to fresh 15 ml conical tubes. During centrifugation, anti-FLAG M2 magnetic beads (SIGMA) were prepared: 25 μl 50%

slurry was aliquoted for each IP (two 150 mm plates), and the beads were washed in batch mode with 3 × 1 ml of lysis buffer. To the rest of the lysate, the equivalent of 12.5 μl packed FLAG M2 magnetic beads was added, and the mixture incubated 2 h at 4 °C with gentle agitation (nutator). Beads were pelleted by centrifugation (1000 rpm for 1 min) and a 15 μl aliquot of the lysate post-IP was taken for analysis. Most of the supernatant was removed with a pipette, and the beads were transferred with ~200 μl of lysis buffer to a fresh 1.7 ml Eppendorf tube, magnetized for ~30 s, and the remaining buffer was aspirated. Two washes with 1 ml lysis buffer and two washes with 20 mM Tris-HCl (pH 8.0) 2 mM CaCl₂ were performed. Briefly, for each of these quick washes, the sample was demagnetized, resuspended by pipetting up and down in the wash buffer, remagnetized for ~30 sec, and the supernatant aspirated (a complete wash cycle takes between 1–2 min). After the last wash, most of the liquid was removed, the tube was spun briefly (1000 rpm for 1 min).

*Tryptic digestion.* Following affinity purification, the beads were resuspended in 5 μl of 20 mM Tris-HCl (pH 8.0). 500 ng of trypsin (Sigma Trypsin Singles, T7575; resuspended at 200 ng/ul in Tris buffer) was added, and the mixture was incubated at 37 °C with agitation for 4 hr. After this first incubation, the sample was magnetized and the supernatant transferred to a fresh tube. Another 500 ng of trypsin was added, and the resulting sample was incubated at 37 °C overnight (no agitation required). The next morning, formic acid was added to the sample to a final concentration of 2% (from a 50% stock solution).

*Mass spectrometry.* Half the sample was used per analysis. A spray tip was formed on fused silica capillary column (0.75 μm ID, 350 μm OD) using a laser puller (program = 4; heat = 280, FIL = 0, VEL = 18, DEL = 200). In all, 10 cm (±1 cm) of C18 reversed-phase material (Reprosil-Pur 120 C18-AQ, 3 μm) was packed in the column by pressure bomb (in MeOH). The column was then pre-equilibrated in buffer A (6 μl) before being connected in-line to a NanoLC-Ultra 2D plus HPLC system (Eksigent, Dublin, USA) coupled to an LTQ-Orbitrap Velos (Thermo Electron, Bremen, Germany) equipped with a nanoelectrospray ion source (Proxeon Biosystems, Odense, Denmark). The LTQ-Orbitrap Velos instrument under Xcalibur 2.0 was operated in the data-dependent mode to automatically switch between MS and up to 10 subsequent MS/MS acquisitions. Buffer A was 100% H₂O, 0.1% formic acid; buffer B was 100 ACN, 0.1% formic acid. The HPLC gradient program delivered the acetonitrile gradient over 125 min. For the first 20 min, the flow rate was of 400 μl/min at 2% B. The flow rate was then reduced to 200 μl/min and the fraction of solvent B increased in a linear fashion to 35% until min 95.5. Solvent B was then increased to 80% over 5 min and maintained at that level until 107 min. The mobile phase was then reduced to 2% B until the end of the run (125 min). The parameters for data-dependent acquisition on the mass spectrometer were: 1 centroid MS (mass range 400–2000) followed by MS/MS on the 10 most abundant ions. General parameters were: activation type = CID, isolation width = 1 m/z, normalized collision energy = 35, activation Q = 0.25, activation time = 10 ms. For data-dependent acquisition, minimum threshold was 500, the repeat count = 1, repeat duration = 30 s, exclusion size list = 500, exclusion duration = 30 s, exclusion mass width (by mass) = low 0.03, high 0.03.

*Mass spectrometry data extraction.* RAW mass spectrometry files were converted to mzXML using ProteoWizard (3.0.4468) and analyzed using the iProphet pipeline[72] implemented within ProHits[73] as follows. The database consisted of the human and adenovirus complements of the RefSeq protein database (version 57) supplemented with "common contaminants" from the Max Planck Institute (http://lotus1.gwdg.de/mpg/mmbc/maxquant_input.nsf/7994124a4298328fc125748d0048fee2/$FILE/contaminants.fasta) and the Global Proteome Machine (GPM; http://www.thegpm.org/crap/index.html). The search database consisted of forward and reversed sequences (labeled "DECOY"); in total 72,226 entries were searched. The search engines used were Mascot (2.3.02; Matrix Science) and Comet[74] (2012.01 rev.3) with trypsin specificity (two missed cleavages were allowed) and deamidation (NQ) and oxidation (M) as variable modifications. Charges of +2, +3, and +4 were allowed, and the parent mass tolerance was set at 12 ppm while the fragment bin tolerance was set at 0.6 amu. The resulting Comet and Mascot search results were individually processed by PeptideProphet[75] and peptides were assembled into proteins using parsimony rules first described in ProteinProphet[76] into a final iProphet protein output using the Trans-Proteomic Pipeline (TPP; Linux version, v0.0 Development trunk rev 0, Build 201303061711). TPP options were as follows: general options were -p0.05 -x20 -PPM -d"DECOY", iProphet options were –ipPRIME and PeptideProphet options were –OpdP. All proteins with a minimal iProphet protein probability of 0.05 were parsed to the relational module of ProHits. Note that for analysis with SAINT, only proteins with iProphet protein probability ≥0.95 are considered. This corresponds to an estimated protein-level FDR of ~5%. Statistical analysis was performed with SAINTexpress (with default parameters), using 38 biological replicates of FLAG-GFP (all from asynchronous HEK293 T-REx cells, all run on the Orbitrap Velos) as negative controls, including two samples run in tandem with the two biological replicates. Full list of high-confidence interaction partners are provided in Supplementary Data 1. The AP-MS data generated in this study have been deposited to the ProteomeXchange database through partner MassIVE under accession codes PXD026809 and MSV000087664, respectively (see Data Availability).

## In vitro peptide- calcineurin-binding assays

*Purification of calcineurin.* 6xHis-tagged human calcineurin A (α isoform, truncated at residue 392), WT or $^{330}$NIR$^{332}$-AAA mutant were expressed in tandem with the calcineurin B subunit in *E. coli* BL21 (DE3) cells (Invitrogen, USA) and cultured in LB medium containing carbenicillin (50 μg/ml) at 37 °C to mid-log phase. Expression was induced with 1 mM IPTG at 16 °C for 18 h. Cells were pelleted, washed, and frozen at −80 °C for at least 12 h. Thawed cell pellets were resuspended in lysis buffer (50 mM Tris-HCl pH 7.5, 150 mM NaCl, 0.1% Tween 20, 1 mM β-mercaptoethanol, protease inhibitors) and lysed by sonication using four, 1-minute pulses at 40% output. Extracts were clarified using two rounds of centrifugation (20,000×*g*, 20 min) and then loaded to 1 ml of Ni-NTA agarose beads (Invitrogen) in lysis buffer containing 5 mM imidazole for 2–4 h. at 4 °C, in batch. Bound beads were loaded onto a column and washed with lysis buffer containing 20 mM imidazole and eluted with lysis buffer containing 300 mM imidazole, pH 7.5. Purified calcineurin heterodimer were dialyzed in buffer (50 mM Tris-HCl pH 7.5, 150 mM NaCl, 1 mM β-mercaptoethanol) and stored in 10% glycerol at −80 °C.

*Peptide purification.* In all, 16mer peptides were fused to GST in vector pGEX-4T-3 and expressed in *E. coli* BL21 (DE3) (Invitrogen). Cells were grown at 37 °C to mid-log phase and induced with 1 mM IPTG for 2 h. Cell lysates were prepared using the EasyLyse™ bacterial protein extract solution (Lucigen Corp. USA) or the CelLytic B reagent (Sigma, USA) according to the manufacturers' protocol and were stored at −80 °C.

*In vitro binding.* In all, 1–2 μg His-tagged calcineurin was first bound to magnetic Dynabeads (Thermo Fisher Sci. USA) in base buffer (50 mM Tris-HCl pH 7.5, 150 mM NaCl, 0.1% Tween 20, 1 mM β-mercaptoethanol, protease inhibitors, 5–10 mM imidazole, 1 mg/ml BSA) for 1 h at 4 °C. In all, 4–5 μg GST-peptide were then added to the binding reaction and incubated further for 3 h. In total, 3% of the reaction mix was removed as 'input' prior to the incubation, boiled in 2X-SDS sample buffer, and stored at −20 °C. The beads were washed in base buffer containing 15–20 mM imidazole and bound proteins were eluted by boiling in SDS sample buffer for 5 min followed by SDS–PAGE and immunoblotting with anti-GST (BioLegend MMS-112P) and anti-His (Qiagen 34660) antibodies. Blots were imaged with the Li-Cor Odyssey imaging system. GST peptides co-purifying with HIS-CN were normalized to their respective input and amount of calcineurin pulled down. Co-purification with CN was reported relative to that of the peptide with the known PxIxIT motif from NFATC1: PALES**SPRIEIT**SCLGL. For (Supplementary Fig. 4B, FAM126A peptides used were FAM126A PSISIT: SGQQRP**PSISIT**LSTD and FAM126A ASASAA Mut: SGQQRP**ASASAA**LSTD. Statistical significance was determined with unpaired Student's *t* test, using GraphPad. Uncropped and unprocessed scans of the blots are provided in the Source Data file.

## Immunoprecipitations

Cells expressing indicated plasmids were rinsed with ice-cold PBS, harvested and pellets snap-frozen in liquid nitrogen and stored at −80 °C until use. Cell pellets were lysed in lysis buffer (50 mM Tris, pH 7.5, 150 mM NaCl, 1% Triton X-100) supplemented with a protease and phosphatase inhibitor cocktail (Halt™, ThermoFisher) and 250 U/ml benzonase for 30 min rotating end-over-end at 4 °C and subjected to fine needle aspiration using a sterile 27.5-gauge needle. Cell lysates were clarified by centrifugation at 16,000×*g* for 20 min and protein concentrations determined by Bicinchoninic acid (BCA) assay. In all, 600–1000 μg of each lysate, adjusted to 1 mg/ml with binding buffer (50 mM Tris, pH 7.5, 150 mM NaCl, 0.5% Triton X-100) was incubated with Pierce anti-HA (ThermoFisher) or GFP-Trap magnetic beads (Bulldog Bio. Inc.) and rotated for 2–4 h at 4 °C. Beads were washed thrice in binding buffer and boiled in 2X SDS sample buffer for 5 min. Input (2%) and bound (100%) fractions were resolved by SDS–PAGE and immunoblotted with HA (1 : 2000, H3663, Sigma-Aldrich), GFP (1 : 4000, Living Colors, 632380, Clontech), MYC (1 : 3000, 9B11, Cell Signaling Technologies), and β-Actin (1:3,000; 926-42210, Li-Cor Biosciences) antibodies followed by secondary Li-Cor antibodies. Blots were imaged with the Li-Cor Odyssey imaging system. Uncropped and unprocessed scans of the blots are provided in the Source Data file.

## Proximity-dependent biotin identification (BioID) analysis

HeLa cells were seeded onto 10 cm plates and transfected at 80% confluence with MYC-BirA-CNAβ1. In all, 24 h post transfection, cells were passaged onto two 10 cm plates. The next day, cells were co-transfected with EFR3B HA_T2A_TTC7B GFP_P2A_FLAG FAM126A (WT or ASASAA mutant) and HA-PI4KA. In all, 4 h post-transfection, media was replaced with fresh media containing 50 μM D-biotin (Sigma B-4501). After 16 h of labeling, cells were collected and snap frozen in liquid nitrogen. Cells were lysed in RIPA buffer (150 mM NaCl, 1% Triton X-100, 0.5% Deoxycholate, 0.1% SDS, 50 mM Tris pH 8.0) supplemented with a protease and phosphatase inhibitor cocktail (Halt™, ThermoFisher) and 250 U/ml benzonase (EMD Millipore) for 30 min rotating end-over-end at 4 °C and subjected to fine needle aspiration with a sterile 27.5-gauge needle. Cell lysates were clarified by centrifugation (16,000 × *g* for 20 min) and protein concentration determined with BCA analysis. For each binding reaction, 1 mg of clarified lysate was incubated with

30 μl of pre-rinsed streptavidin magnetic particles (11641786001, Sigma-Aldrich) in 1 ml RIPA buffer for 16 h with rotation at 4 °C. An input aliquot (20 μl) was removed prior to bead addition. Beads were washed thrice with 1 ml RIPA buffer, rotating for 5 min each, and eluted in 2X sample buffer (10%SDS, 0.06% Bromophenol blue, 50% glycerol, 0.6 M DTT, 375 mM Tris-HCl pH 6.8). Inputs and bound (100%) samples were boiled and resolved by SDS–PAGE followed by immunoblotting with mouse FLAG (1:2,500; F3165, Sigma- Aldrich), rabbit MYC (1:2,000; 71D10, Cell Signaling), mouse HA (1 : 2000, H3663, Sigma-Aldrich), mouse GFP (1:4,000, Living Colors, 632380, Clontech), and rabbit β-Actin (1 : 3000; 926-42210, Li-Cor Biosciences) antibodies. Blots were imaged with Li-Cor Odyssey imaging system following incubation with secondary Li-Cor antibodies. Binding for each protein was quantified as their respective signals/MYC bound signal normalized to respective signals/Actin Input signal. Biotinylation of each protein in complex with WT FAM126A, is set to 1. Statistical significance was determined using GraphPad.

## Hydrogen-deuterium exchange analysis (HDX-MS)

*Protein expression.* GST-tagged human calcineurin A (residues 2-391 of human CNA alpha isoform) in tandem with calcineurin B subunit were expressed in BL21 C41 *Escherichia coli*, induced with 0.1 mM IPTG (isopropyl β-d-1-thiogalactopyranoside) and grown at 23 °C overnight. Cells were harvested, flash frozen in liquid nitrogen, and stored at −80 °C until use. Bacmids harboring MultiBac PI4KA complex constructs were transfected into *Spodoptera frugiperda* (Sf9) cells, and viral stocks amplified for one generation to acquire a P2 generation final viral stock. Final viral stocks were added to Sf9 cells at ~1.8 × 10$^6$ in a 1/100 to 1/50 virus volume to cell volume ratio. Constructs were expressed for 68 h before pelleting of infected cells. Cell pellets were snap frozen in liquid nitrogen, followed by storage at −80 °C.

*Protein purification (GST tagged human calcineurin).* *Escherichia coli* cell pellets were lysed by sonication for 5 min in lysis buffer [50 mM Tris pH 8.0, 100 mM NaCl, 2 mM EDTA, 2 mM EGTA, protease inhibitors (Millipore Protease Inhibitor Cocktail Set III, Animal-Free)]. NaCl solution was added to 1 M and the solution was centrifuged for 10 min at 12,000×*g* at 1 °C and for 20 min at 38,000×*g* at 1 °C (Beckman Coulter Avanti J-25I, JA 25.50 rotor). CHAPS was added to 0.02%. Supernatant was loaded onto a 5 ml GSTrap 4B column (GE) in a superloop for 45 min and the column was washed in Wash Buffer [50 mM Tris pH 8.0, 110 mM KOAc, 2 mM MgOAc, 1 mM DTT, 5% glycerol (v/v), 0.02% chaps] to remove nonspecifically bound proteins. The column was washed in Wash Buffer containing 2 mM ATP to remove the GroEL chaperone. The GST-tag was cleaved by adding Wash Buffer containing PreScission protease to the column and incubating overnight at 4 °C. Cleaved protein was eluted with Wash Buffer. Protein was concentrated using an Amicon 10 kDa MWCO concentrator (MilliporeSigma) and size exclusion chromatography (SEC) was performed using a Superdex 75 10/300 column equilibrated in Wash Buffer. Fractions containing protein of interest were pooled, concentrated, flash frozen, and stored at − 80 °C.

*Protein purification (PI4KA complex).* Sf9 pellets were resuspended in lysis buffer [20 mM imidazole pH 8.0, 100 mM NaCl, 5% glycerol, 2 mM βMe, protease (Protease Inhibitor Cocktail Set III, Sigma)] and lysed by sonication. Triton X-100 was added to 0.1% final, and lysate was centrifuged for 45 min at 20,000×*g* at 1 °C. (Beckman Coulter Avanti J-25I, JA 25.50 rotor). Supernatant was loaded onto a HisTrap FF Crude column (GE Healthcare) and superlooped for 1 h. The column was washed with Ni-NTA A buffer [20 mM imidazole pH 8.0, 100 mM NaCl, 5% glycerol (v/v), 2 mM βMe], washed with 6% Ni-NTA B buffer [30 mM imidazole pH 8.0, 100 mM NaCl, 5% (v/v) glycerol, 2 mM βMe], and the protein eluted with 100% Ni-NTA B buffer (450 mM imidazole). Elution fractions were passed through a 5 ml StrepTrapHP column pre-equilibrated in GF buffer [20 mM imidazole pH 7.0, 150 mM NaCl, 5% glycerol (v/v), 0.5 mM TCEP]. The column was washed with GF buffer before loading a tobacco etch virus protease containing a stabilizing lipoyl domain (Lip-TEV), and cleavage proceeded overnight. Cleaved protein was eluted with GF buffer and concentrated down to 250 μl in an Amicon 50 kDa MWCO concentrator (MilliporeSigma) pre-equilibrated in GF buffer. Concentrated protein was flash frozen in liquid nitrogen and stored at −80 °C.

*Mass spectrometry sample preparation.* HDX reactions for PI4KA complex (PI4KIIIα, TTC7B, FAM126A) and Calcineurin were conducted in a final reaction volume of 24 μl with a final concentration of 0.17 μM (8 pmol) PI4KA complex and 0.95 μM (24 pmol) Calcineurin. The reaction was initiated by the addition of 16 μl of D$_2$O buffer (150 mM NaCl, 20 mM pH 8.0 Imidazole, 90% D$_2$O (V/V)) to 6.5 μl of PI4KA or PI4K buffer solution and 0.66 μl Calcineurin or Calcineurin buffer solution (final D$_2$O concentration of 65%). The reaction proceeded for 3, 30, 300, or 3000 s at 20 °C, before being quenched with ice-cold acidic quench buffer, resulting in a final concentration of 0.6 M guanidine-HCl and 0.9% formic acid post quench. All conditions and timepoints were generated in triplicate. Samples were flash frozen immediately after quenching and stored at −80 °C until injected onto the ultra-performance liquid chromatography (UPLC) system for proteolytic cleavage, peptide separation, and injection onto a QTOF for mass analysis, described below.

*Protein digestion and MS/MS data collection.* Protein samples were rapidly thawed and injected onto an integrated fluidics system containing a HDx-3 PAL liquid handling robot and climate-controlled (2 °C) chromatography system (LEAP Technologies), a Dionex Ultimate 3000 UHPLC system, as well as an Impact HD QTOF Mass spectrometer (Bruker). The full details of the automated LC system are described in[77]. The protein was run over one immobilized pepsin column (Trajan; ProDx protease column, 2.1 mm × 30 mm PDX.PP01-F32) at 200 μL/min for 3 min at 10 °C. The resulting peptides were collected and desalted on a C18 trap column (Acquity UPLC BEH C18 1.7 mm column (2.1 × 5 mm); Waters 186003975). The trap was subsequently eluted in line with an ACQUITY 1.7 μm particle, 100 × 1 mm² C18 UPLC column (Waters), using a gradient of 3–35% B (Buffer A 0.1% formic acid; Buffer B 100% acetonitrile) over 11 min immediately followed by a gradient of 35–80% over 5 min. Mass spectrometry experiments acquired over a mass range from 150 to 2200 m/z using an electrospray ionization source operated at a temperature of 200 °C and a spray voltage of 4.5 kV.

*Peptide identification.* Peptides were identified from the non-deuterated samples of PI4K using data-dependent acquisition following tandem MS/MS experiments (0.5 s precursor scan from 150 to 2000 m/z; 12 0.25 s fragment scans from 150 to 2000 m/z). MS/MS datasets were analyzed using PEAKS7 (PEAKS), and peptide identification was carried out by using a false discovery-based approach, with a threshold set to 1% using a database of known contaminants found in Sf9 cells and BL21 C41 *Escherichia coli*[78]. The search parameters were set with a precursor tolerance of 20 ppm, fragment mass error 0.02 Da, charge states from 1 to 8, leading to a selection criterion of peptides that had a −10logP score of 35.4 and 29.3 for the PI4KA complex and calcineurin, respectively. A search for phosphorylated peptides was subsequently completed using PEAKS7 with a variable phosphorylation modification search with the same parameters as above, and a false discovery threshold of 0.1% using a database of known contaminants. The search parameters were otherwise the same as above leading to a selection criterion of peptides that had a −10logP score of 37.8 for the PI4KA complex. All putative phosphorylated peptides from the search were subjected to manual inspection, and the only one site (TTC7B S519) was identified. No phosphorylation of FAM126A was detected. The area of the phosphorylated and unphosphorylated peptides of TTC7 S159 were roughly equal, suggesting the purified trimer was ~50% phosphorylated at this site.

*Mass analysis of peptide centroids and measurement of deuterium incorporation.* HD-Examiner Software (Sierra Analytics) was used to calculate the level of deuterium incorporation into each peptide. All peptides were manually inspected for correct charge state, correct retention time, and appropriate selection of isotopic distribution. Deuteration levels were calculated using the centroid of the experimental isotope clusters. Results are presented as relative levels of deuterium incorporation, with no correction for back exchange. The only correction was for the deuterium percentage of the buffer in the exchange reaction (65%). Differences in exchange in a peptide were considered significant if they met all three of the following criteria: ≥5% change in exchange, ≥0.5 Da difference in exchange, and a two-tailed T-test value of less than 0.01. The raw HDX data are shown in two different formats. The raw peptide deuterium incorporation graphs for a selection of peptides with significant differences are shown in Fig. 4e, g, with the raw data for all analyzed peptides in the Source Data file. To allow for visualization of differences across all peptides, we utilized number of deuteron difference (#D) plots (Fig. 4d). These plots show the total difference in deuterium incorporation over the entire H/D exchange time course, with each point indicating a single peptide. Samples were only compared within a single experiment and were never compared to experiments completed at a different time with a different final D₂O level. The data analysis statistics for all HDX-MS experiments are in Supplementary Table 1 according to published guidelines[79]. The HDX-MS data generated in this study have been deposited to the ProteomeXchange database through partner PRIDE[80] under accession code PXD025900.

**In vivo analysis of FAM126A phosphorylation status.** HeLa cells seeded onto 10 cm plates were transfected with EFR3B HA_T2A_ TTC7B MYC_P2A_FLAG FAM126A WT or ASASAA mutant. In all, 24 h post-transfection, plates were washed, trypsinized, and passaged onto 60 mm plates for treatments. The next day, cells were pre-treated with 2 μM FK506 (LC Laboratories) for 1 h, 2 μM BIM (bisindolylmaleimide, Calbiochem) for 15 min or DMSO in growth media. Cells were then stimulated with 500 nM PMA (Phorbol 12-myristate 13-acetate, Sigma-Aldrich) or DMSO for 15 min, washed and harvested in ice-cold PBS. Pellets were snap-frozen in liquid nitrogen and stored at −80 °C until use. Cells were lysed with RIPA buffer (150 mM NaCl, 1% Triton X-100, 0.5% Deoxycholate, 0.1% SDS, 50 mM Tris pH 8.0) supplemented with protease and phosphatase inhibitor cocktail (Halt™, ThermoFisher) and 250 U/ml benzonase for 30 min rotating end-over-end at 4 °C and subjected to fine needle aspiration with a sterile 27.5-gauge needle. Cell lysates were clarified by centrifugation at 16,000 × g for 20 min) and protein concentration determines using BCA analysis. In all, 20 μg of each lysate was analyzed by SDS–PAGE followed by immunoblotting with anti-FLAG (1 : 1000) and custom anti-phosphospecific FAM126A S485 (1 : 1000) antibodies. PKC activation was assessed by phosphorylation of the downstream substrate, ERK using p44/42 Erk1/2 antibody (1 : 3000; 3A7, Cell Signaling Technologies). Phosphorylation status of FAM126A in each treatment was quantified as pFAM126A

S485 signal/FLAG signal and reported relative to that in DMSO-treated FAM126A WT sample, using ImageStudio imaging software. Statistical significance was determined using GraphPad. Uncropped and unprocessed scans of the blots are provided in the Source Data file.

**Bioluminescence resonance energy transfer measurements.** HEK 293T cells were trypsinized and plated on white 96-well plates at a density of 75,000 cells/100 μl per well, together with the indicated DNA constructs (0.15 μg total DNA in 25 μl per well) and the cell transfection reagent (1.5 μl GeneCellin (Bulldog Bio) in 25 μl per well) in Opti-MEM reduced serum medium (Gibco). Cells were transfected with DNA encoding the human M3 muscarinic receptor (0.1 μg total DNA/well) and the previously established SidM-2XP4M-based PI4P biosensor[22] (0.05 μg total DNA/well). After 6 h, media were replaced with 100 μl/well of DMEM supplemented with 10% fetal bovine serum, 50 U/ml penicillin, and 50 μg/ml streptomycin. Measurements were performed 28 h post-transfection. Prior to measurements, media was replaced with 50 μl buffer containing 120 mM NaCl, 4.7 mM KCl, 1.2 mM CaCl₂, 0.7 mM MgSO₄, 10 mM glucose, and 10 mM Na-HEPES, pH 7.4. Cells were pretreated with FK506 (1 μM), Cyclosporin A (10 μM), or DMSO for 1 h at 37 °C. Measurements were performed at 37 °C using a Varioskan Flash Reader (Thermo Scientific) and initiated with the addition of the cell permeable luciferase substrate, coelenterazine h (20 μl, final concentration of 5 μM). Counts were recorded using 485 and 530 nm emission filters. Detection time was 250 ms for each wavelength. The indicated reagents were also dissolved in modified Krebs-Ringer buffer and were added manually in 10 μl. For this, plates were unloaded, which resulted in an interruption in the recordings. All measurements were done at least in triplicate. BRET ratios were calculated by dividing the 530 nm and 485 nm intensities and results were normalized to the baseline. Since the absolute initial ratio values depended on the expression of the sensor, the resting levels were considered as 100%, whereas the 0% was determined from values of those experiments where cytoplasmic Renilla luciferase construct was expressed alone.

**Statistical analysis.** Statistics were computed using Graphpad Prism 9. All data shown as representative images or as the mean of measurements with standard deviation (SD) error bars unless noted otherwise. All data represent at least three independent experiments and are indicated in each figure legend. For image analysis in Fig. 1f and Supplementary Fig. 1f, number of cells analyzed for GM130 co-localization from three independent experiments were as follows: $n = 166$ for CNAβ2, $n = 164$ for CNAβ1, $n = 130$ for CNAβ1$^{C483}$, $n = 128$ for CNAβ1$^{C493S}$, $n = 119$ for CNAβ1$^{C2S}$. For plasma membrane signal ratio measurements: $n = 75$ for CNAβ2, $n = 86$ for CNAβ1, $n = 98$ for CNAβ1$^{C483}$, $n = 80$ for CNAβ1$^{C493S}$, $n = 77$ for CNAβ1$^{C2S}$ from three independent experiments. Number of cells used for image analysis in Fig. 2g from 4 independent experiments are $n = 76$ for vector control, $n = 94$ for wildtype ABHD17A, $n = 89$ for ABHD17A S190A mutant. Figure 1g immunoblot is representative of $n = 5$ for EFR3B-FLAG, $n = 5$ for FLAG-CNAβ1, $n = 3$ for each CNAβ1 mutant (C483S, C493S, C2S), all replicates were independent experiments. Two-tailed unpaired Student's $t$-test was applied for statistical analyses between two groups. One-way analysis of variance (ANOVA) with appropriate multiple comparisons (all indicated in figure legends) were performed when comparing more than two groups.

**Reporting summary.** Further information on research design is available in the Nature Research Reporting Summary linked to this article.

## Data availability
The data that support this study are available from the corresponding author upon reasonable request. The AP-MS data generated in this study have been deposited to the ProteomeXchange database through partner MassIVE under accession codes PXD026809 and MSV000087664, respectively. The HDX-MS data generated in this study have been deposited to the ProteomeXchange database through partner PRIDE under accession code PXD025900. Uncropped and unprocessed scans of blots and quantifications as well as HDX-MS source data are provided in the Source Data file. The imaging data that was used to generate Figs. 1e, f and 2f, g and Supplementary Fig. 1 e, f are published in Mendeley Data with DOI: 10.17632/85v4dj4kgm.1[81]. Structures of calcineurin heterodimer and the PI4KA trimer used in Fig. 4 are obtained from the Protein Data Bank with accession codes 6NUC and 6BQ1 respectively. Source data are provided with this article. Source data are provided with this paper.

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

## Acknowledgements

We thank Callie Preast Wigington for critical reading of the manuscript, Rachel Bond for generating most of the CNAβ plasmids used in this study and for initiating experiments to identify CNAβ-interacting proteins, Pin-Joe Ko for helpful discussions about quantitative image analysis, Jamin Hein for assistance in developing the phospho-specific FAM126A antibody, the Skotheim lab for cell lines, the De Camilli lab for plasmids and all members of the Cyert lab for their support and critical feedback. M.S.C., J.R. and I.U.-T. are supported by grants from the National Institute of Health R01GM129236 and R35GM136243. J.E.B. acknowledges funding support from a Discovery grant from the Natural Science and Engineering Research Council of Canada (NSERC-2020-04241, J.E.B.), and the Michael Smith Foundation for Health Research (J.E.B., Scholar Award 17686). G.G. and T.B. are supported by the Intramural Research Program of the Eunice Kennedy Shriver National Institute of Child Health and Human Development of the NIH. A.C.G. is supported by a grant from the Canadian Institutes of Health Research (FDN 143301). P.V. is supported by the Hungarian National Research, Development, and Innovation Fund (NKFIK134357). E.C. acknowledges support from the Canadian Institutes of Health Research (PJT-162184).

## Author contributions

I.U.-T. and M.S.C. jointly designed the study. I.U.-T. performed the majority of the experiments and analyzed data, supervised by M.S.C., with the exception of the following: M.A.H.P., M.L.J. and J.E.B. designed and executed HDX-MS experiments. J.R. carried out in vitro binding studies. A.Z.L.S. and E.C. designed and performed pulse-chase analyses of palmitate incorporation. N.St-D. and A.-C.G. designed and conducted AP-MS experiments. BRET-based analyses of PI4P dynamics were designed and executed by P.V. (effects of CN inhibitors) and by G.G. and T.B. (analyses of FAM126A mutants). I.U.-T. and M.S.C. wrote the manuscript with editorial input and approval from all authors.

## Competing interests

The authors declare no competing interests.
