## [Peer Review File · Nature Communications]

Palmitoylation targets the Calcineurin phosphatase to the Phosphatidylinositol 4-kinase complex at the plasma membraneReviewers' Comments:

Reviewer #1:

Remarks to the Author:

This work explores the understudied isoform of calcineurin CNA β 1. It was found that CNA β 1 is recruited to the membrane in a palmitoylation-dependent manner, that the palmitoylation state is dynamic, and that ABHD17A is responsible for demaplitoylation of CNA β 1. PI4KA is identified as a specific interactor of CNA β 1 above CNA β 2, and that this interaction occurs via the PSISIT motif at the C-terminus of FAM126A. This interaction was investigated via affinity purifications, proximity-dependent labelling, and hydrogen-deuterium exchange-mass spectrometry. FAM126A was found to be a substrate of CN dephosphorylation, and inhibitors of calcineurin affect the production of PI4P by PI4KA.

In my opinion this research is extremely high-quality. The data are convincing, with good controls in place for all experiments. The conclusions are backed up well by the data, which is of high quality and presented well in the figures. Whilst I don't have expert knowledge of calcineurin, the work appears to be of high significance as described by the authors. The paper is well-written and well-structured, and I recommend it for publication in Nature Communications.

Some minor points for clarification are as follows-

Line 53-55; Not clear what the abbreviations stand for. It would be clearer to state 'Calcineurin (CN), also sometimes known as...'

The bottom part of figure 1e should be separate from the top and middle part, because the cells have been treated differently. The bottom panels have been treated with 17-ODYA, and it's slightly confusing to refer back to the same figure in a later section of the manuscript.

To increase the understandability for readers who aren't familiar with the pulse-chase experiments, they should be better described, with more references (e.g. Martin and Cravatt Nat. Prot. 2009 and Zhang et al. PNAS 2009). For example, why is the L-AHA used, and what role does the click-chemistry play.

Line 325- HDX also reports protection of protein regions from the solvent due to protein-protein interactions, as well as secondary structure content.

Line 419- heterotrimer is spelled incorrectly.

In Figure 4 e&G, error bars should be added to the graphs.

Line 876- If the protein probability is >0.95, this is an estimated protein-level FDR of 5%, not 0.5%?

Figure 4d- From which timepoint(s) is this data taken?

Reviewer #2:

Remarks to the Author:

In their manuscript entitled "Palmitoylation targets the Calcineurin phosphatase to the Phosphatidylinositol 4-kinase complex at the plasma membrane" Ulengin-Talkish et al., examine an understudied calcineurin isoform, CNA β 1 and show that CNA β 1 localizes to the plasma membrane and Golgi due to palmitoylation of its divergent C-terminal tail and that this reaction is reversed by the ABHD17A depalmitoylase. The authors further show that palmitoylation targets CNA β 1 to a distinct set of membrane-associated interactors including the phosphatidylinositol 4-kinase (PI4KA) complex

containing EFR3B, PI4KA, TTC7B and FAM126A. The application of calcineurin inhibitors decrease PI4P production during Gq-coupled GPCR signaling, suggesting that calcineurin dephosphorylates and promotes PI4KA complex activity. In sum, this work discovers a new calcineurin-regulated signaling pathway highlighting the PI4KA complex as a regulatory target and revealing that dynamic palmitoylation confers unique localization, substrate specificity and regulation to CNA β 1. Overall this is a nice compact study, with potential interest to the scientists working on the particular field. Most of the experiment are adequately described and performed. Many different analysis methods are used, for which I complement the authors. Several moderate-to-minor issues however should be resolved prior to publication.

Comments & issues:

-75% of the commercially available cell lines used in this study are gifts from other labs. The authors state in the reporting summary that the authentication of these are in progress. The authentication results would be good to have prior to publication.

-It is somewhat unorthodox to refer to manuscript figures in the introduction (Fig 1a-b). I think this part would be better suited to the results section.

-Why does the cis-Golgi marker GM130 look different in the Fig. 1e (CNA β 2 compared to CNA β 1)? This could effect the calculations on the Fig. 1F.

-The HEK293 T-REx cell lines developed for AP-MS analyses should be checked that the expression levels of the 3X FLAG-tagged CNA β 2 and CNA β 1 match to the endogenous expression levels of these proteins. Overexpression of the tagged-transgenes can often results in overexpression artefacts.

-the authors employ BioID approach in the manuscript, however, it would be highly important to perform the BioID MS experiments minimally with CNA β 2, CNA β 1 and CNA β Δ , and to compare the results with the AP-MS results.

-It is not clear how the 12 interacting proteins were selected for the Fig3b out of >50 interactions. This should be explained clearly.

Reviewer #3:

Remarks to the Author:

In this manuscript, Ulengin-Talkish et al. investigate the function of CNA β 1, an isoform of the CNA subunit of the calcium-regulated phosphatase calcineurin. This splice form is poorly studied and uniquely localizes to the plasma membrane. The authors show that this localization is dependent on the palmitoylation of two highly conserved cysteine residues specific to this isoform. Furthermore, by using a proteomics approach, the authors show that FAM126A, EFR3B, TTC7B and PI4KA, all components of the PI4KA lipid kinase complex, are all enriched with CNA β 1 IP. The direct interaction between CNA β 1 and FAM126A is validated by identifying a CNA β 1-binding motif on FAM126A and confirmed by performing HDX-MS experiments. The authors go on to show that FAM126A is phosphorylated on conserved serine residues, and that the CNA β 1 isoform of calcineurin can specifically dephosphorylate S485. The authors finally show that inhibiting this dephosphorylation event reduces the increase in PI4KA-mediated PI4P production occurring upon GPCR-Gq signaling, providing some evidence that this calcium-regulated dephosphorylation event may be functionally relevant to the activity of PI4KA and phosphoinositide homeostasis at the plasma membrane. Overall, this is a rigorous study using an impressive breadth of cutting-edge molecular and cellular approaches. It is also timely in that it provides a compelling mechanistic model to explain aspects of how cells regulate an enzyme complex that controls an important step of lipid biosynthesis. I enthusiastically recommend publication in Nat Commun after the authors address the minor comments

below.

1. Fig 2b-c: If I understand correctly, palmostatin treatment increases palmitoylation of the AHA-pulse-labeled fraction of calcineurin over two hours of chase. Based on the imaging and acyl-PEG assay, it would seem that most/all of CNA β 1 is palmitoylated. Wouldn't the palmostatin/ODYA/AHA experiment suggest that the fraction of it that is palmitoylation increases during this time? Or because it reports on the fraction palmitoylated with ODYA (as opposed to regular palmitate), it is reporting on the half-life of palmitoylation? Space-permitting, the authors may want to comment more on the interpretation of these experiments.

2. Related to point 1: the acyl-PEG assay in Fig 1g shows substantial non-palmitoylated form of CNA β 1 (and EFR3B), but imaging of these from this manuscript and for EFR3B in previous studies show them entirely on membranes. The authors may want to comment on the potential "partial false negative" results from acyl-PEG assays, i.e., reduced antibody labeling of PEG-labeled proteins, incomplete reaction during the lengthy protocol, etc.

3. Fig 2f-g: Could the authors comment on why overexpression of ABHD17 removes the plasma membrane but not Golgi localization of calcineurin? ABHD17 has a punctate/intracellular localization, including what looks like the Golgi complex, so it should be in the right place to depalmitoylate the Golgi-localized protein too, in principle.

4. Fig 3d-e the selective enrichment of β 1-WT vs. β 1-C2S or β 2 in the EFR3B-HA IP is modest (\sim 2-fold) compared to what would be expected from the AP-MS experiments. Are the interactions of β 2 or β 1-C2S with EFR3B-HA "real" or background? Figure 4b clearly shows that the interaction depends upon FAM126A, so this is more of a technical question than one that would alter the key conclusions of the paper.

5. Why were HDX experiments not performed in the presence of membranes? Ideally EFR3B would be a part of these studies as well, though recombinant expression of the full-length protein has not been demonstrated.

6. The results in Fig 5 do not provide a mechanism by which FAM126A S485 is phosphorylated, as only PKC was tested. Presumably the authors have considered alternate hypotheses, and though determination of kinases that control particular phosphorylation sites can be very challenging, it would help to at least have a discussion of some proposed possibilities. If the dephosphorylation is acute/calcium-stimulated, would the phosphorylation be expected to be tonic and occurring in non-stimulated cells? Based on the data in Fig S5a showing that the phosphorylation of FAM126A, even when it is in complex with PI4KA and TTC7B, can only occur when it is tethered to the plasma membrane by EFR3B, such kinase activity is likely at the plasma membrane. A discussion of some of these issues would be helpful.

7. The BRET data provide good suggestive evidence for the proposed mechanism of CNA β 1 in regulating PI4KA activity by FAM126A S485 dephosphorylation. Though additional phosphorylation of the complex by PKC at other sites may be responsible for the full effect, one prediction of the calcineurin-dependent portion of this model is that the in vitro kinase activity assay of PI4KA complexes containing TTC7B and either WT or ASASAA FAM126A would have different catalytic activities. It would strengthen the conclusions if these studies could be performed (as the Burke group has performed in Dornan et al., JMB 2018). The statement about overexpression and data not shown suggests that studies of this type have been attempted in cells. If this has not been attempted in vitro, it may help to support the model for the role of S485 phosphorylation.

8. The authors may want to mention that AF647-azide is the click chemistry partner for 17-ODYA within the text and gel images in the figures, not just the methods section, for clarity of presentation.

9. Typos: ABHD17A (p.9) and FAM126A (p.12 before HDX section)

REVIEWER COMMENTS

Reviewer #1 (Remarks to the Author):

This work explores the understudied isoform of calcineurin CNA β 1. It was found that CNA β 1 is recruited to the membrane in a palmitoylation-dependent manner, that the palmitoylation state is dynamic, and that ABHD17A is responsible for depalmitoylation of CNA β 1. PI4KA is identified as a specific interactor of CNA β 1 above CNA β 2, and that this interaction occurs via the PSISIT motif at the C-terminus of FAM126A. This interaction was investigated via affinity purifications, proximity-dependent labelling, and hydrogen-deuterium exchange-mass spectrometry. FAM126A was found to be a substrate of CN dephosphorylation, and inhibitors of calcineurin affect the production of PI4P by PI4KA.

In my opinion this research is extremely high-quality. The data are convincing, with good controls in place for all experiments. The conclusions are backed up well by the data, which is of high quality and presented well in the figures. Whilst I don't have expert knowledge of calcineurin, the work appears to be of high significance as described by the authors. The paper is well-written and well-structured, and I recommend it for publication in Nature Communications.

We thank the reviewer for their positive assessment of the quality and significance of our work. We have addressed their specific comments in the resubmitted revised manuscript as below:

Some minor points for clarification are as follows-

-Line 53-55; Not clear what the abbreviations stand for. It would be clearer to state 'Calcineurin (CN), also sometimes known as....'

As suggested by the reviewer, the revised manuscript now states, (line 53) "Calcineurin (CN), also known as PP2B or PPP3, is the conserved Ca²⁺/calmodulin (CaM)-activated serine/threonine protein phosphatase.....".

-The bottom part of figure 1e should be separate from the top and middle part, because the cells have been treated differently. The bottom panels have been treated with 17-ODYA, and it's slightly confusing to refer back to the same figure in a later section of the manuscript.

We apologize to the reviewer about this confusion. However, the bottom part of Figure 1e does not show cells that have been treated with 17-ODYA. Rather, these are COS-7 cells that are expressing CNA β 1 (C483,493S) in which serine has been substituted for 2 cysteines that are sites of palmitoylation. This mutant thus fails to be palmitoylated and displays cytosolic localization. None of the cell images shown in the manuscript are from cell treated with 17-ODYA and this figure is not referred to again in a later section. Was the reviewer asking about images in Figure 2? These show effects of ABHD17A overexpression (WT and

catalytically inactive) on CNA β 1 distribution, but again no 17-ODYA was used in these analyses.

-To increase the understandability for readers who aren't familiar with the pulse-chase experiments, they should be better described, with more references (e.g. Martin and Cravatt Nat. Prot. 2009 and Zhang et al. PNAS 2009). For example, why is the L-AHA used, and what role does the click-chemistry play.

We thank the reviewer for this comment and note that reviewer 3 also had questions about this experiment. We added the references suggested by the reviewer and re-wrote this section to better explain the experimental setup and the use of dual labelling to simultaneously monitor palmitate dynamics while controlling for protein turnover. We hope that this new text (lines 206-218) has been sufficiently clarified.

*-Line 325- HDX also reports protection of protein regions from the solvent due to protein-protein interactions, as well as secondary structure content. We thank the reviewer for this comment. While extensive experimentation has defined the primary role of secondary structure stability as the main determinant of H/D exchange, we agree that changes can also occur due to solvent exclusion and have changed the wording to mention this fact. In the revised manuscript this sentence reads, "HDX-MS measures the exchange rate of amide hydrogens with deuterium-containing buffer, which acts as a sensitive probe of secondary structure dynamics **and solvent accessibility**."*

-Line 419- heterotrimer is spelled incorrectly. Thank you, this typo has been fixed.

*-In Figure 4 e&G, error bars should be added to the graphs. Thank you for this comment. In fact, all of the panels do include error bars, however many of these are smaller than the size of the point and are therefore not visible. We modified the legend for Figure 4 to reflect this: "**All error bars in panels d-g show S.D. (n = 3), with many being smaller than the size of the point.**"*

-Line 876- If the protein probability is >0.95, this is an estimated protein-level FDR of 5%, not 0.5%? We thank the reviewer for catching this error. The sentence, in a section that we moved to supplementary methods in the revised manuscript, now states that the estimated protein-level FDR is 5%.

*-Figure 4d- From which timepoint(s) is this data taken? We thank the reviewer for this question. The differences in HDX shown in Figure 4 are from the biggest difference at any timepoint. The full raw data (Supplementary table 3) includes all differences with and without PI4KA / Calcineurin. We have added more details in the methods and figure legend to explain this. The modified legend for Figure 4d specifically now reads, "d. **Sum of the differences** in the number of deuterium incorporation for all analyzed peptides over the deuterium exchange (HDX) time course."*

Reviewer #2 (Remarks to the Author):

In their manuscript entitled "Palmitoylation targets the Calcineurin phosphatase to the Phosphatidylinositol 4-kinase complex at the plasma membrane" Ulengin-Talkish et al., examine an understudied calcineurin isoform, CNAb1 and show that CNAb1 localizes to the plasma membrane and Golgi due to palmitoylation of its divergent C-terminal tail and that this reaction is reversed by the ABHD17A depalmitoylase. The authors further show that palmitoylation targets CNAb1 to a distinct set of membrane-associated interactors including the phosphatidylinositol 4-kinase (PI4KA) complex containing EFR3B, PI4KA, TTC7B and FAM126A. The application of calcineurin inhibitors decrease PI4P production during Gq-coupled GPCR signaling, suggesting that calcineurin dephosphorylates and promotes PI4KA complex activity. In sum, this work discovers a new calcineurin-regulated signaling pathway highlighting the PI4KA complex as a regulatory target and revealing that dynamic palmitoylation confers unique localization, substrate specificity and regulation to CNAb1. Overall this is a nice compact study, with potential interest to the scientists working on the particular field.

Most of the experiment are adequately described and performed. Many different analysis methods are used, for which I compliment the authors. Several moderate-to-minor issues however should be resolved prior to publication.

We thank the reviewer for their appreciation of our work and the breadth of analytical methods used. Responses to specific points follows.

Comments & issues:

-75% of the commercially available cell lines used in this study are gifts from other labs. The authors state in the reporting summary that the authentication of these are in progress. The authentication results would be good to have prior to publication. All of the cell lines have now been authenticated using STR profiling. This is reported in the Eukaryotic Cell lines section of the Nature reporting summary PDF. The raw data showing the results of our authentication analyses is included in the source data file.

-It is somewhat unorthodox to refer to manuscript figures in the introduction (Fig 1a-b). I think this part would be better suited to the results section. We thank the reviewer for this comment. While we agree that this is somewhat unconventional, we would like to use these figures, which present previously published information, to introduce CNAβ1 to readers because the vast majority of them will never have heard of this isoform. While not typical, we have been told that this arrangement is acceptable to the editor and have therefore opted to retain references to these figure panels in the introduction.

-Why does the cis-Golgi marker GM130 look different in the Fig. 1e (CNAβ2 compared to CNAβ1)? This could effect the calculations on the Fig. 1F. We thank the reviewer for noting

this difference in GM130 staining which reflects cell to cell variation and does **not** reflect a true difference in Golgi morphology for CNA β 2 vs CNA β 1 expressing cells. We did not want readers to similarly have this impression. Therefore, in the revised Figure 1e, we replaced the image with one that looks more typical. All of the raw cell images for this figure have been included in a published data set on Mendeley (DOI: 10.17632/85v4dj4kgm.1), so the reviewer (and readers) can confirm that no consistent differences in GM130 staining were observed in CNA β 2-expressing cells.

-The HEK293 T-REx cell lines developed for AP-MS analyses should be checked that the expression levels of the 3X FLAG-tagged CNA β 2 and CNA β 1 match to the endogenous expression levels of these proteins. Overexpression of the tagged-transgenes can often results in overexpression artefacts. We thank the reviewer for this comment and we appreciate their concerns. Interaction proteomics should always be considered only as a useful starting point to examine protein function, and while we cannot completely rule out the possibility that higher expression of the CNA β baits described here identified some erroneous interactions, we have also found previously that overexpression helps capture low-affinity, PxlIT-motif mediated interactions (See Goldman et al (2014) Mol. Cell 55:422-435). Furthermore, the AP-MS analyses presented did identify 12 known CN interactors, and interaction with the PI4KA complex (EFR3B, FAM126A, TTC7B and PI4KA), which is the main focus of this study, was validated using multiple methods.

-The AP-MS analyses in this manuscript utilize a pipeline that was developed to systematically identify interactors for 140 phosphatases (See St-Denis et al (2016), Phenotypic and Interaction Profiling of the Human Phosphatases Identifies Diverse Mitotic Regulators, Cell Reports 17:2488). As stated in that paper, the expression system employed (HEK293 T-Rex FlipIn cell lines) "has been successfully used in the past to achieve moderate expression levels for signaling molecules (e.g. Glatter et al., 2009; Lambert et al., 2013)". In St-Denis et al (2016), expression levels were directly assessed for six baits including PPP3CA, and were either slightly lower or roughly similar for five of them as shown in Figure S1B of that paper. Thus, this approach has been broadly used to identify many physiologically relevant interactions.

- In the revised manuscript, we have included immunoblot analyses of bait expression levels cell lines used for our AP-MS analyses as **Supplementary Figure 3b**. This shows that of the three FLAG-tagged CNA β baits analyzed, (CNA β 1, CNA β 2 and CNA β trunc), CNA β 1, the focus of this study, is expressed at the lowest level; spectral counts were not normalized for bait expression level. Unfortunately, commercial antibodies that are specific for CNA β isoforms are not sensitive enough to identify endogenous protein levels in these extracts, which prevents us from making a direct comparison with transgene levels.

-We have added the following to the results section of the revised manuscript, "This system has been successfully used to achieve moderate expression levels for signaling proteins and identify biologically relevant interactors for other protein phosphatases (St-Denis et al 2016, Glatter et al. 2009, Lambert et al., 2013), although we were unable to directly compare

expression levels of the transgenes with endogenous proteins due to the low sensitivity of CNA β -specific antibodies.”

-the authors employ BioID approach in the manuscript, however, it would be highly important to perform the BioID MS experiments minimally with CNA β 2, CNA β 1 and CNA β Δ , and to compare the results with the AP-MS results. We completely agree with the reviewer that BioID-MS analyses would provide complementary information to AP-MS analyses and that this approach is better able to capture low-affinity interactions. We have published such analyses with CNA α (Wigington et al (2020), Mol. Cell 79:342) and have proposed such experiments with CNA β 1 in a recent grant proposal. However, carrying out these analyses represents a significant undertaking. Because the focus of the current work is on characterizing the identified interaction with the PI4KA complex rather than presenting a comprehensive analysis of CNA β 1 interactors, we respectfully suggest that these experiments are beyond the scope of this paper and better left for future studies.

-It is not clear how the 12 interacting proteins were selected for the Fig3b out of >50 interactions. This should be explained clearly.

We apologize to the reviewer for this confusion. We added this information to the results section: “Excitingly, several proteins preferentially associated with CNA β 1 relative to CNA β 2 or CNA β trunc, i.e. spectral counts \geq 1.5x more for CNA β 1 than other baits (Fig. 3b).” It is also stated in the legend for Figure 3.

Reviewer #3 (Remarks to the Author):

In this manuscript, Ulengin-Talkish et al. investigate the function of CNA β 1, an isoform of the CNA subunit of the calcium-regulated phosphatase calcineurin. This splice form is poorly studied and uniquely localizes to the plasma membrane. The authors show that this localization is dependent on the palmitoylation of two highly conserved cysteine residues specific to this isoform. Furthermore, by using a proteomics approach, the authors show that FAM126A, EFR3B, TTC7B and PI4KA, all components of the PI4KA lipid kinase complex, are all enriched with CNA β 1 IP. The direct interaction between CNA β 1 and FAM126A is validated by identifying a CNA β 1-binding motif on FAM126A and confirmed by performing HDX-MS experiments. The authors go on to show that FAM126A is phosphorylated on conserved serine residues, and that the CNA β 1 isoform of calcineurin can specifically dephosphorylate S485. The authors finally show that inhibiting this dephosphorylation event reduces the increase in PI4KA-mediated PI4P production occurring upon GPCR-Gq signaling, providing some evidence that this calcium-regulated dephosphorylation event may be functionally relevant to the activity of PI4KA and phosphoinositide homeostasis at the plasma membrane. Overall, this is a rigorous study using an impressive breadth of cutting-edge molecular and cellular approaches. It is also timely in that it provides a compelling mechanistic model to explain aspects of how cells regulate an enzyme complex that controls an important step of lipid biosynthesis. I enthusiastically recommend publication in Nat Commun after the authors

address the minor comments below.

We thank the reviewer for their enthusiastic endorsement of our study and their appreciation for its timeliness and significance. Specific responses follow.

1. Fig 2b-c: If I understand correctly, palmostatin treatment increases palmitoylation of the AHA-pulse-labeled fraction of calcineurin over two hours of chase. Based on the imaging and acyl-PEG assay, it would seem that most/all of CNA β 1 is palmitoylated. Wouldn't the palmostatin/ODYA/AHA experiment suggest that the fraction of it that is palmitoylation increases during this time? Or because it reports on the fraction palmitoylated with ODYA (as opposed to regular palmitate), it is reporting on the half-life of palmitoylation? Space-permitting, the authors may want to comment more on the interpretation of these experiments. We thank the reviewer for pointing out that we need to explain this somewhat complicated experiment more clearly. Reviewer 1 also requested further explanation of this section, which has been expanded and clarified in the revised manuscript: see. Lines 206-218. In short, the second statement by the reviewer is correct—that is ODYA labelling of the L-AHA labelled pool of CNA β 1 turns over during the chase period via depalmitoylation and repalmitoylation with non ODYA-labelled palmitate. Thus, inclusion of PalmB increases the relative ODYA labelling of this L-AHA-labelled pool by preventing replacement of 17-ODYA with unlabelled palmitate during the chase period.

2. Related to point 1: the acyl-PEG assay in Fig 1g shows substantial non-palmitoylated form of CNA β 1 (and EFR3B), but imaging of these from this manuscript and for EFR3B in previous studies show them entirely on membranes. The authors may want to comment on the potential "partial false negative" results from acyl-PEG assays, i.e., reduced antibody labeling of PEG-labeled proteins, incomplete reaction during the lengthy protocol, etc. We thank the reviewer for pointing out this apparent contradiction. In the revised manuscript we cite a reference about APE that points out some of its limitations (Percher et al. (2017) Curr Protocol Protein Sci). Partial false negatives can occur during APE because of insufficient thioester hydrolysis by hydroxylamine, or inefficient labeling with PEG. Thus, in terms of reporting the extent of palmitoylation, APE is only semi-quantitative, but is useful for directly comparing samples (such as in point #3 below). Indeed in our own experiments we always see 2 pegylated forms for CNA β 1, but variable amounts of non-shifted protein.

We have added the following sentence to the manuscript (lines 172-173):

"Mass-tag conversion may be incomplete during APE; thus this method accurately reports S-acylation states, but not the fraction of protein in the sample that is modified (Percher (2017))."

3. Fig 2f-g: Could the authors comment on why overexpression of ABHD17 removes the plasma membrane but not Golgi localization of calcineurin? ABHD17 has a punctate/intracellular localization, including what looks like the Golgi complex, so it should be in the right place to depalmitoylate the Golgi-localized protein too, in principle. Include some

text?: We thank the reviewer for this question. In the revised manuscript we included a new data panel, Supplemental Figure 2a, in which APE analysis was carried out on extracts of cells co-expressing CNA β 1 and ABHD17A (WT or catalytically impaired.) The results indicate that in ABHD17A overexpressing extracts, both shifted forms of CNA β 1 are much reduced, and especially the dually shifted band. The revised manuscript includes new text (Lines 230-234): "Furthermore, APE analysis of these samples showed that compared to vector and ABHD17A(S190A), expression of ABHD17A WT significantly reduced palmitoylation of GFP-CNA β 1, especially the dually palmitoylated form that is required for stable PM association. This is consistent with CNA β 1 localization to the Golgi in these cells (Supplementary Figure 2a)."

-Also, please note that we added a new reference to a recent paper from the Cravatt lab about ABDH substrate specificity (Line 493-97 in discussion): "Here we show that a membrane-localized thioesterase, ABHD17A, which regulates H- and N-RAS⁴⁴, also catalyzes the depalmitoylation of CNA β 1 causing it to redistribute from the PM to the cytosol and the Golgi. **Recent work indicates that the ABHD17 family of depalmitoylases specifically targets PM-associated proteins (Remsberg et al. (2021) Nat. Chem Biol.)**, although mechanisms that control the activity of these proteins are yet to be identified.

4. *Fig 3d-e the selective enrichment of β 1-WT vs. β 1-C2S or β 2 in the EFR3B-HA IP is modest (~2-fold) compared to what would be expected from the AP-MS experiments. Are the interactions of β 2 or β 1-C2S with EFR3B-HA "real" or background? Figure 4b clearly shows that the interaction depends upon FAM126A, so this is more of a technical question than one that would alter the key conclusions of the paper. We thank the reviewer for this question. In short, we believe that both results are 'correct' but represent differences in expression levels of PI4KA components in the two analyses. For both experiments (AP-MS and EFR3B co-IPs), CNA β transgenes have been integrated into HEK293 T-REX cells via the FLiP-In method, and are thus likely expressed at modestly elevated levels (See response to reviewer 2). However, in AP-MS experiments, all of the prey proteins including EFR3B, TTC7B, FAM126A and PI4KA are endogenously expressed, and can only identified after being effectively solubilized digested and detected via mass spectrometry. These conditions yielded low numbers of peptides for PI4KA complex members (See Supplemental table 1). By contrast, for the co-IP experiments presented in Figure 3d-e, PI4KA components were epitope-tagged and expressed using the plasmids we created to co-express balanced levels of EFR3B, TTC7B and FAM126A along with PI4KA which was co-transfected on a separate plasmid. Therefore, the detected interactions w; EFR3B are likely 'real' for experiments shown in Figure 3d-e, but reflect these higher expression levels of PI4KA complex components relative to calcineurin.*

5. *Why were HDX experiments not performed in the presence of membranes? Ideally EFR3B would be a part of these studies as well, though recombinant expression of the full-length protein has not been demonstrated.*

The addition of membranes into HDX experiments greatly complicates their analysis. While we (Burke lab) have extensively characterized PI3Ks on membranes (~200 kDa), the much larger size of the PI4KA heterotrimer (~450 kDa) greatly complicates this work. Also, as noted by the reviewer, the main determinant in driving membrane recruitment is the lipidated EFR3 protein. The Burke lab has made extensive attempts to generate lipidated EFR3 in Sf9 cells which have not yet succeeded. While we agree that this is an important goal, due to the extensive complications, we believe these analyses are best left for future experiments."

6. The results in Fig 5 do not provide a mechanism by which FAM126A S485 is phosphorylated, as only PKC was tested. Presumably the authors have considered alternate hypotheses, and though determination of kinases that control particular phosphorylation sites can be very challenging, it would help to at least have a discussion of some proposed possibilities. If the dephosphorylation is acute/calcium-stimulated, would the phosphorylation be expected to be tonic and occurring in non-stimulated cells? Based on the data in Fig S5a showing that the phosphorylation of FAM126A, even when it is in complex with PI4KA and TTC7B, can only occur when it is tethered to the plasma membrane by EFR3B, such kinase activity is likely at the plasma membrane. A discussion of some of these issues would be helpful.

We thank the reviewer for these questions. We agree that identifying the kinases that phosphorylate FAM126A and other PI4KA complex members is a critical future goal. Unfortunately, the few kinase inhibitors we tried did not affect S485 phosphorylation and computational analysis of this site using NetPhos 3.1 was not able to predict a kinase for this site. We do note, in agreement with the reviewer, that the kinase(s) responsible must be active under basal conditions as the phospho-specific antibody recognizes FAM126A under basal conditions, and that it must be PM associated as FAM126A is only phosphorylated when expressed with the membrane tether for the PI4KA complex, EFR3B. To address the reviewer's request to discuss some of these issues in the manuscript, we have added the following sections of text in the revision:

-Line 372: "These slower migrating forms, indicative of hyperphosphorylation, suggest that that FAM126A is phosphorylated only when associated with the PM-localized PI4KA complex **i.e. by a PM-associated protein kinase**, and that CN dephosphorylates FAM126A in a PxlIT-dependent manner."

-Lines 404-405: "Furthermore, addition of PMA did not enhance S485 phosphorylation **which was detected under basal conditions, suggesting that this site is constitutively phosphorylated**".

-Discussion, lines 496-504, "Our results provide the first insights into this regulation by demonstrating that CN binds to and modulates the phosphorylation of at least one site in the FAM126A tail (Ser485) in cells, which, based on our findings, is phosphorylated by an unidentified kinase that is active at the PM under basal signaling conditions. Computational analysis failed to predict any likely candidate kinases for this site (NetPhos 3.1), however up

to 10% of human kinases localize to the PM, including many that are uncharacterized (Zhang et al. (2021) eLife 10:e64943). Further studies are required not only to identify relevant kinases, but also to comprehensively map CN-regulated phosphosites in PI4KA complex members, and assess the functional consequences of these modifications.”

7. *The BRET data provide good suggestive evidence for the proposed mechanism of CNAB1 in regulating PI4KA activity by FAM126A S485 dephosphorylation. Though additional phosphorylation of the complex by PKC at other sites may be responsible for the full effect, one prediction of the calcineurin-dependent portion of this model is that the in vitro kinase activity assay of PI4KA complexes containing TTC7B and either WT or ASASAA FAM126A would have different catalytic activities. It would strengthen the conclusions if these studies could be performed (as the Burke group has performed in Dornan et al., JMB 2018). The statement about overexpression and data not shown suggests that studies of this type have been attempted in cells. If this has not been attempted in vitro, it may help to support the model for the role of S485 phosphorylation. We completely agree with the reviewer that analyses of PI4K activity would be an ideal addition to this work. However, if we were to express the WT or ASASAA FAM126A containing complexes in SF9 cells, unfortunately the complexes would not be produced with the appropriate phosphorylated residues, so this would not allow the phosphorylated/dephosphorylated states to be compared. Currently, we do not know which kinase(s) are responsible for modification of S485 and/or any additional phosphorylation sites in FAM126A or other PI4KA complex components that contribute to regulation. In the revised manuscript we modified the methods section to report on our comprehensive characterization of the phosphorylation state of the WT PI4KA heterotrimer purified from SF9 cells, where no phosphorylation at S485 is observed. This is not surprising, as maintaining phosphorylated proteins during purification can be challenging, and requires extensive use of phosphatase inhibitors, and activators of phosphorylation. Due to the extreme challenges likely inherent in generation of stoichiometrically phosphorylated S485 PI4KA trimer, and lack of information about regulatory sites in other PI4KA complex members, we feel that these experiments are best suited to future studies.*

8. *The authors may want to mention that AF647-azide is the click chemistry partner for 17-ODYA within the text and gel images in the figures, not just the methods section, for clarify of presentation. We thank the reviewer for these suggestions. We have added the click chemistry partners for 17-ODYA and L-AHA to the text of the results and have added click-chemistry partners for these analogs to the labels for gel images in all the relevant figures.*

9. *Typos: ABHD17A (p.9) and FAM126A (p.12 before HDX section). Thank you. These typos have been fixed.*